



# BIOPERIANT12: a mesoscale resolving coupled physics-biogeochemical model for the Southern Ocean

Nicolette Chang[1,2], Sarah-Anne Nicholson[1], Marcel du Plessis[3], Alice D. Lebehot[1], Thulwaneng Mashifane[1], Tumelo C. Moalusi[1,2], N. Precious Mongwe[1,4], and Pedro M.S. Monteiro[1,2,5]

[1]Southern Ocean Carbon-Climate Observatory, CSIR, Cape Town, South Africa
[2]Global Change Institute, University of the Witwatersrand, Johannesburg, South Africa
[3]Department of Marine Science, University of Gothenburg, Sweden
[4]National Institute for Theoretical and Computational Sciences (NITheCS), South Africa
[5]School for Climate Studies, Stellenbosch University, Stellenbosch, South Africa

**Correspondence:** Nicolette Chang (nchang@csir.co.za)

**Abstract.** We present BIOPERIANT12, a regional model configuration of the Southern Ocean (SO) at a mesoscale-resolving $1/12°$. This is a stable, ocean–ice–biogeochemical configuration derived from the Nucleus for European Modelling of the Ocean (NEMO) modelling platform. It is specifically designed to investigate questions related to the mean state, seasonal cycle variability and mesoscale processes in the mixed layer and within the upper ocean (<1000 m). In particular, the focus is on
understanding processes behind carbon and heat exchange, systematic errors in biogeochemistry and assumptions underlying the parameters chosen to represent these SO processes. The dynamics of the ocean model play a large role in driving ocean biogeochemistry and we show that over the chosen period of analysis 2000–2009 that the simulated dynamics in the upper ocean provide a stable mean state, as compared to observation-based datasets (themselves subject to biases such as sparsity of data, cloud cover, etc.), and through which the characteristics of variability can be described. Using ocean biomes to delineate
the major regions of the SO, the model demonstrates a useful representation of ocean biogeochemistry and partial pressure of carbon dioxide ($pCO_2$). In addition to a reasonable model mean state performance, through model–data metrics BIOPERIANT12 highlights several pathways for improving Southern Ocean model simulations such as the representation of temporal variability and the overestimation of biological biomass.



# 1   Introduction

The Southern Ocean (SO) is an essential sink of carbon dioxide ($CO_2$) and heat, responsible for nearly 40 % of the global
ocean annual mean $CO_2$ uptake and 75 % of the excess heat globally (Frölicher et al., 2015; Gruber et al., 2023). Since
the 2000s, this sparsely observed region has been increasingly sampled (Williams et al., 2018; Meredith et al., 2013; Swart
et al., 2012). These efforts, conducted at increasing temporal and spatial resolutions, have exposed the challenge of accurately
representing $CO_2$ heat fluxes of the SO within ocean models and by extension, earth system models (ESMs) from which

projections are made (Rintoul, 2018), despite being weakly constrained and seasonally biased themselves. The seasonal cycle
is arguably the dominant mode of variability in physical-biogeochemical properties of the SO (Lenton et al., 2013; Thomalla
et al., 2011; Mongwe et al., 2018; Gregor et al., 2019; Rodgers et al., 2023), yet the state-of-the-art, particularly the ocean
components of ESMs (Chassignet et al., 2020; Treguier et al., 2023), display inadequate representation of these scales. Large
model biases in ESMs are seen in the wide, inter-model spread of the contributing models in the previous generations of

Climate Model Intercomparison Project (CMIP). This is particularly evident in the pronounced inability to reproduce similar
seasonal cycles of air-sea $CO_2$ flux ($FCO_2$) (Anav et al., 2013; Lenton et al., 2013; Kessler and Tjiputra, 2016; Mongwe et al.,
2016, 2018), sea–ice extent and trends (Meijers, 2014; Beadling et al., 2020), mixed layer depth (MLD) (Sallée et al., 2013;
Treguier et al., 2023), water mass properties (Downes et al., 2015; Beadling et al., 2020), dissolved iron (Tagliabue et al., 2016)
and phytoplankton phenology (Thomalla et al., 2011, 2023; Hague and Vichi, 2018). Key to reproducing the heat and $CO_2$

fluxes in the SO, are the representation of the dynamics in models, particularly the ubiquity of mesoscale eddies due to the
strong baroclinically induced instabilities of the Antarctic Circumpolar Current (ACC) and associated strong mesoscale kinetic
energy (Daniault and Ménard, 1985; Smith et al., 2023). Mesoscale dynamics explain a substantial portion of the annual and
seasonal variance in mixed-layer depths (Whitt et al., 2019; Gaube et al., 2019), and therefore influence global circulation by
impacting water mass transformation and influencing the SO overturning circulation through eddy compensation and sensitivity

of wind (Abernathey et al., 2016; Munday et al., 2014). Mesoscale and also submesoscale dynamics, through enhancements
in advection and mixing, consequently influence the local biogeochemistry (BGC) and thus carbon exchange, such as by
altering the supply of limiting nutrients to the euphotic layer (Frenger et al., 2015; Nicholson et al., 2019; Uchida et al., 2019)
and available light via "eddy slumping" induced stratification during spring (Lévy et al., 1998, 1999; Marshall et al., 2002;
Lévy et al., 2010; Mahadevan et al., 2012). Simulations with coupled physics-BGC in global (Rohr et al., 2020) and regional

ocean models (Song et al., 2018; Uchida et al., 2019, 2020) show the variability in both biological and physical mechanisms
due to eddies influence the biological response in the SO with implications both spatially and temporally, such as in the
characteristics of the seasonal cycles of both physical and BGC processes. Computational cost is still a limiting factor in the
design of both projection and process models; while computational power (and horizontal model resolution) has increased from
CMIP5 (resolutions of 1° or coarser and relied on suitable parameterisations), CMIP6 only has resolutions of up 0.25° which

allows some explicit representation of the ocean (Hewitt et al., 2020; Haarsma et al., 2016), these do not include the mesoscale.
Model configurations with BGC, therefore, compromise in spatial/temporal resolution or run duration and model complexity.
For example, the MOMSO configuration (Modular Ocean Model Southern Ocean, Dietze et al., 2020) is eddy-resolving in the





SO (11 km resolution at 40° S) but has a reduced BGC model (BLING) consisting of 4 BGC tracers and a carbon module for
climatologically forced, multi-decadal experiments. SO iron supply experiments by Uchida et al. (2019), on the other hand,
use a full BGC model with phytoplankton and zooplankton functional groups, but an idealised ocean of 20 km resolution, flat-
bottomed, re-entrant channel. BGC data assimilation for SO flux experiments in B-SOSE (Biogeochemical Southern Ocean
State Estimate) uses model resolution (1/3°, Verdy and Mazloff, 2017, and 1/6°, http://sose.ucsd.edu/) and run duration which
relies on periods of sufficient observations. These configuration compromises contribute to uncertainties in carbon and heat
exchange, particularly for climate-scale questions using long-running models (Hewitt et al., 2020; Beadling et al., 2020). The
importance of model resolution is apparent through, firstly, the representation of ocean dynamics that distribute BGC tracers in
the model and are the major driver of the BGC differences of simulations with observations and between simulations in ESMs,
for example, mesoscale modulations of the MLD affect light and iron supply to surface influencing phytoplankton growth
and thus the biological carbon pump (Song et al., 2018). Secondly, resolution affects the evolution of the tracers themselves,
(Séférian et al., 2013) discuss errors in the BGC fields which may propagate spatially and amplify during model evolution in
low resolution simulations, particularly the pools of nutrients and iron which supply the surface, although this matters more for
simulations of hundreds of years versus the short model duration of BIOPERIANT12. To understand the heat and carbon biases
in SO models, we focus on understanding the biophysical dynamics in the ocean surface mixed-layer, the boundary across
which atmosphere-ocean exchange occurs. In this paper, we present a regional, circumpolar, mesoscale-resolving (1/12°),
contemporary ocean–ice–biogeochemical, NEMO–PISCES model configuration, BIOPERIANT12. This configuration has a
laterally unconstrained ACC with resolved eddies, and a prescribed atmosphere to evolve the SO over a limited duration (20
years). We describe the model design in Section 2. In Section 3, we assess the configuration looking at model stability and,
through a comparison against observations, suitability to serve as an experimental platform and for downscaling, submesoscale
experiments and sensitivity studies coinciding with in situ experiments (Swart et al., 2012; du Plessis et al., 2022; Djeutchouang
et al., 2022; Smith et al., 2023).

## 2 BIOPERIANT12 model configuration

BIOPERIANT12 (full configuration name: BIOPERIANT12-CNCLNG01) is a regional, mesoscale-resolving configuration for
simulating the ocean, ice, and biogeochemistry of the circumpolar SO for contemporary conditions. NEMO-PISCES version
3.4 (Gurvan et al., 2019) with specifications by the DRAKKAR consortium (Barnier et al., 2014) was used. This consists of an
ocean-ice component provided by the OPA (Océan Parallélisé) model and Louvain-la-Neuve Sea Ice Model (LIM2) and bio-
geochemical components by PISCES (Aumont and Bopp, 2006; updated version Aumont et al., 2015). Although more recent
versions were available at production, this model version was used to be consistent with configuration development, starting
with a hierarchy of tests of increasing resolution on which model parameters were tested. This model run was configured from
two other previous model runs from the DRAKKAR Group and will be referred to several times below: ORCA12-MAL101,
a global physics only run (Barnier et al., 2014; Lecointre et al., 2011) and BIOPERIANT05-GAA95b, an eddy-permitting SO





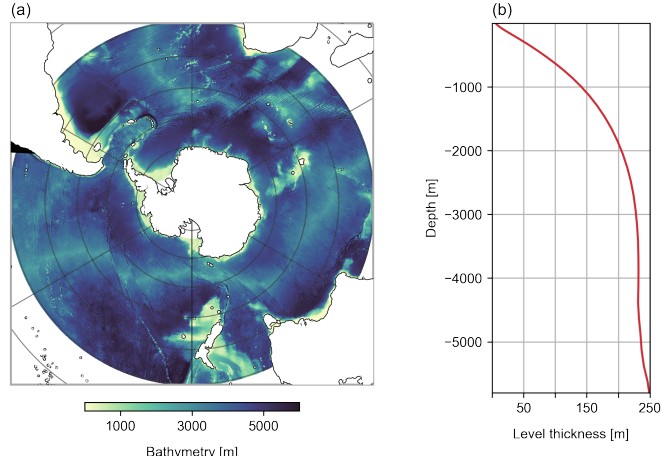

**Figure 1.** BIOPERIANT12 grid configuration **(a)** domain showing model bathymetry and **(b)** vertical grid thickness versus depth.

configuration including biogeochemistry (Albert, A. pers. comm, MEOM-DRAKKAR group), an updated model of Dufour et al. (2013) from which we derived our designation of BIOPERIANT12.

## 2.1 Domain and grid

The model grid and bathymetry for the Southern Ocean south of 30° S (Fig. 1a) is a subset of the global tripolar ORCA12 grid with 46 vertical levels built from the ETOPO2 dataset combined with the GEBCO one minute grid. The horizontal grid, at

1/12° resolution (∼8 km at 30° S, 4.6 km at 60° S), can be considered eddy-resolving. The vertical grid of BIOPERIANT12 consists of 46 z-coordinate levels with partial bottom steps. For the surface ocean mixed layer fluxes and biogeochemistry, there are 15/18 levels in the upper 200/400 m, with vertical resolution ranging from 6–40 m. Below this, grid thickness increases: ∼200 m at 2000 m depth (z = 29) and finally 250 m at the bottom (Fig. 1b).

## 2.2 Initial conditions

To obtain a representative and stable surface ocean state, BIOPERIANT12 is initialised from rest and for tracers from respective climatological products. For the ocean, Levitus temperature and salinity climatology (Locarnini et al., 2010) were used. Sea ice initial conditions are from the January ice climatology for 1998–2007 from ORCA12-MAL101, when sea ice is at a minimum in the seasonal cycle and thereafter freely evolves with the LIM2 model. Biogeochemical tracers were initialised from coarse-resolution climatologies: dissolved inorganic carbon (DIC), total alkalinity (TA) from the GLODAP annual climatology (Key

et al., 2004); nitrate ($NO_3$), phosphate ($PO_4$), silicate (Si) and oxygen ($O_2$) from the January monthly WOA09 climatology (Garcia et al., 2013, 2010); dissolved organic carbon and iron were consistent with the BIOPERIANT05-GAA95b initialisation 90 from an ORCA05 run. The other tracers were initialised with constant values set in PISCES.





## 2.3 Boundary Conditions

BIOPERIANT12 has one open lateral boundary in the north. Boundary conditions for the dynamics were derived from an
ORCA12–MAL101 from the available 5 day averages for 1989–2009. While the BIOPERIANT12 dynamical open boundary
conditions are of a comparative 1/12° resolution, biogeochemical boundary conditions were not available at this high resolution
at the time of preprocessing; we therefore obtained them from the coarser 1/2° BIOPERIANT05-GAA95b. A climatological
"normal year" boundary forcing consisting of 5 day averages for the period 1995–2009 was calculated and used as opposed to
the interannual dataset due to reported biases in certain biogeochemical tracers (Mongwe et al., 2016). For the lateral boundary
of the coastlines partial free slip is prescribed. Surface atmospheric forcing data are from ERA-interim (Dee et al., 2011) using
the CORE bulk formulation. Wind components are supplied at 3 hourly intervals using the absolute wind formulation thus
neglecting effects of the surface ocean current on the wind. Sea surface restoring of salinity to Levitus monthly climatology is
applied. Restoring for Antarctic Bottom Water is applied to counteract the drift in ACC transport due to the representation of
deep waters in the DRAKKAR models (Dufour et al., 2012).

## 2.4 Model evolution

The model is run for the duration 1989 to 2009, the same period for which the ORCA12 boundary conditions were available.
The model integration was carried out with a baroclinic time step of 360 s. Within the first 5 years of initialisation surface ocean
dynamics reach statistical equilibrium and transport through the Drake Passage stabilises, years 1989 to 1994 are therefore
designated as spin-up (Fig. S1, S2). Thereafter the model is run to 2009 with output saved as 5 day averages and the final 10
years (2000–2009) were analysed. At depths below $400\ m$, temperature shows a significant drift from 2002, while the aim is
to simulate surface ocean processes, this must be noted when using the model output.

## 2.5 Model Numerics

Ocean tracers (temperature and salinity) advection is implemented with the TVD scheme, while passive PISCES tracers are
advected using the MUSCL advection scheme. Lateral diffusion of both ocean and passive tracers is achieved with a laplacian
operator along isonetural surfaces. Lateral advection of momentum uses a leapfrog scheme; momentum diffusion uses the
bilaplacian operator along geopotential surfaces. Vertical mixing in the model uses the turbulent kinetic energy (TKE) closure
scheme. For subgrid-scale mixing in the background, vertical eddy viscosity and diffusivity coefficients are set to 1.2 x $10^{-4}$
$m^2\ s^{-1}$ and 1.2 x $10^{-5}\ m^2\ s^{-1}$. At the bottom, a diffusive bottom boundary layer scheme is used as well as the advective bottom
boundary layer scheme for the case of dense water overlying less dense water at the bottom (Gurvan et al., 2019). The bottom
boundary is set with nonlinear bottom friction.

## 2.6 Computational Requirements

Development of BIOPERIANT12 started after the introduction of the Lengau cluster of NICIS-CHPC (Centre for High Per-
formance Computing) in late 2016, which comprises of approximately 32 832 Intel XEON CPU cores when fully operational





and is a shared HPC resource of the South African research community. The final iteration, this reference simulation, was run in 2020. Due to South African electricity restrictions, compute capacity, resource allocation and system stability decrease, to compensate for this, we chose to use more storage space to create and save restart files more frequently. Even with the NEMO land elimination algorithm eliminating 19 % of the non-ocean subdomains, 3240 CPUs were chosen to balance model scaling and wall clock time for running and saving ocean, ice and BGC 5 day outputs as well as restart files.

## 3 Model Evaluation

We evaluate BIOPERIANT12 physical and biogeochemical fields by comparing key metrics of the upper-ocean for the last 10 years of the experiment, 2000–2009, against observations (OBS), for which temporal and spatial coverage improves within the 2000s. Observational datasets are provided inline and summarised in Table 1. We note that many datasets are low resolution, gridded products, which is only applicable in evaluating the large-scale mean-state. We check that the model reproduces annual and seasonal mean states as an initial comparison and then follow this with the characteristics of temporal variability. In Section 3.1, we evaluate the model physical ocean and ice properties as an indication of model stability and the general circulation affecting the BGC tracers, guided by the metrics proposed by Russell et al. (2018) for the evaluation of the SO in coupled climate models and ESMs. As a summary, preceding further evaluation of the BGC output, we present biomes and summarisw the model output compared to that of observations (Section 3.2). In Section 3.3, we present modelled carbon. Finally, biogeochemical and biological properties are further analysed in Section 3.4.

### 3.1 Key physical ocean metrics in the Southern Ocean

#### 3.1.1 Transport through Drake Passage

Transport of the ACC through the Drake Passage at 69° W in BIOPERIANT12 is stable after spin-up (Fig. S2), with annual mean transport through the Drake Passage from 2000–2009 at 145.25 ±5.66 Sv. The value is comparable with observations as summarised in Table 2, such as the generally accepted observational estimate of 134 ±11.2 Sv by Whitworth and Peterson (1985); although estimates of transport taken after 2007 yield values higher than this, e.g 173.3 ±10.7 Sv (Donohue et al., 2016), attributed to higher resolution observations and is used as the observational benchmark for the model analysis of CMIP3 to CMIP6 models by Beadling et al. (2020). It also compares well to its similar SO regional model "predecessors" PERIANT05 and PERIANT025 of (0.5° and 0.25° resolution, respectively), with respective mean transports of 149.2 Sv and 143.2 Sv (Dufour et al., 2012), as well as the multi-model mean of 155 ±51 Sv for CMIP5 global models of mostly 1° resolution (Meijers, 2014) and similar to eddy-resolving Ocean Model Intercomparison Project phase 2 (OMIP-2) models that fall within the chosen observation range of 134–173 Sv (Chassignet et al., 2020).





**Table 1.** Summary of observational datasets used for model evaluation

| Variable | Temporal resolution and coverage | Horizontal resolution | Dataset and reference URL |
|---|---|---|---|
| Currents u, v | 2000–2009 | 0.25° | AVISO altimetry (The Ssalto/Duacs altimeter products were produced and distributed by the Copernicus Marine and Environment Monitoring Service, CMEMS, http://www.marine.copernicus.eu ) |
| Temperature, Salinity | Monthly climatology | 0.25° | WOA2013 (Locarnini et al., 2010) |
| Polar Front position | Weekly 2002–2009 | 0.25° | Satellite (Freeman and Lovenduski, 2016) |
| MLD gridded | Monthly climatology | 1° | Argo profiles (Holte et al., 2017) |
| T/S profiles | 2002–2009 | | Argo profiles (Holte et al., 2017) |
| Sea ice concentration | 2000–2009 | 0.25° | NOAA/NSIDC (Meier et al., 2017; Peng et al., 2013) |
| Mean biomes | Mean over 1998–2010 | 1° | Biome dataset (Fay and McKinley, 2014) |
| FCO$_2$, pCO$_2$ | Monthly means | 1° | CSIR-ML6 multi-platform machine-learning product (Gregor et al., 2019) |
| DIC, TA | Annual mean centred 2002 | 1° | GLODAPv2 (Olsen et al., 2016; Lauvset et al., 2016) |
| Dissolved iron | Binned into months | 1° bins | Dissolved iron in situ profiles (Tagliabue et al., 2014) |
| NO$_3$, PO$_4$, Si | Monthly climatology | 1° | WOA13 (Garcia et al., 2010) |
| Dissolved oxygen | Monthly climatology | 1° | WOA13 (Garcia et al., 2013) |
| Chloropyll-a | Weekly 2000–2009 | 9 km | OC-CCI-v6 mixed satellite in situ chlorophyll-a (Sathyendranath et al., 2019) |

### 3.1.2  Eddy Kinetic Energy (EKE)

Figure 2 shows the climatological annual mean surface EKE for 2000–2009 for the model compared to the EKE derived from AVISO 1/4° gridded altimetry dataset. The distribution of EKE in the model is in general agreement with observations, regions
160 of high EKE such as those associated with western boundary currents and downstream of topography are represented in the model. Zonally averaged EKE bands are comparable to SO models by Munday et al. (2021, Fig. 7a), although BIOPERIANT12 shows slightly higher EKE than their models, except for the WBC regions. BIOPERIANT12 appears to overestimate EKE except between 36 and 43° S (Fig. 2a) particularly from the Agulhas Current region (Fig. S2) as seen in the meridionally averaged EKE between 15–45° E (Fig. S3). Models at similar resolutions are able to represent the distribution patterns of





**Table 2.** BIOPERIANT12 Drake Passage transport vs selected estimates from literature

| Transport [Sv] | Source | Reference |
|---|---|---|
| 145.25 ±5.66 | BIOPERIANT12 | |
| 134 ±11.2 | Observations | Whitworth and Peterson (1985) |
| 173.3 ±10.7 | Observations after 2007 | Donohue et al. (2016) |
| 149.2 | Model 0.5° resolution (PERIANT05) | Dufour et al. (2012) |
| 143.2 | Model 0.25° resolution (PERIANT025) | Dufour et al. (2012) |
| 155 ±51 | CMIP5 multi-model mean (1° resolution) | Meijers (2014) |
| 134–173 | OMIP–2 models eddy-resolving | Chassignet et al. (2020) |

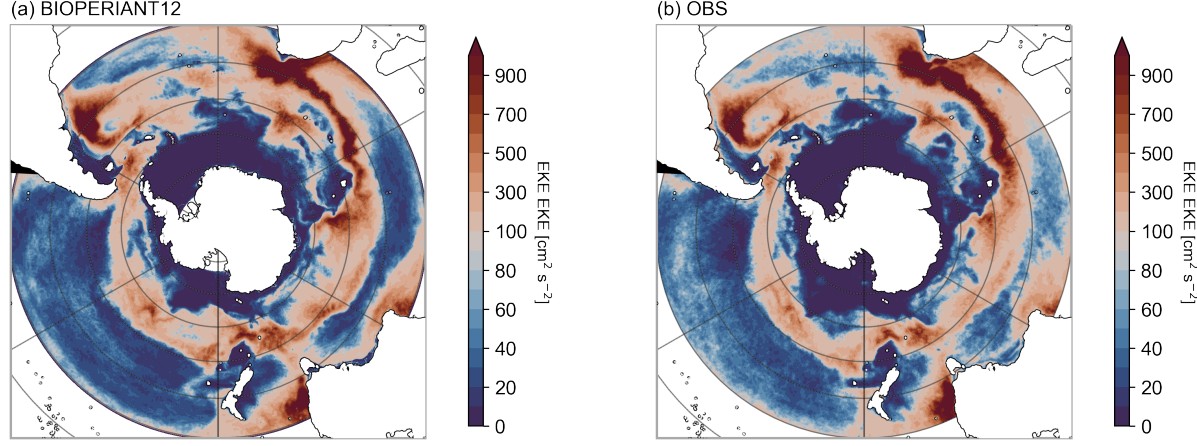

**Figure 2.** Annual mean surface EKE for years 2000–2009 for **(a)** BIOPERIANT12 model and **(b)** the AVISO 1/4° dataset.

EKE (e.g. ORCA12, (Rieck et al., 2015; Patara et al., 2016)), although magnitudes may differ. Most OMIP-2 ocean models (at ~1/10°) underestimate EKE with some contribution from temporal averaging (Chassignet et al., 2020), regional effects may also contribute: high EKE regions are eddy-rich but their spatial coverage in OMIP-2 models are smaller than observations (AVISO) while eddy-poor regions have greater spatial coverage. Regional model MOMSO (Dietze et al., 2020), on the other hand, overestimates EKE, which is attributed to the higher model than observational dataset resolution, degrading model spatial resolution obtains more comparable magnitudes. With consideration to factors such as regional differences, such as in the effect of winds (Patara et al., 2016) and applying the absolute wind formula which neglects the effect of current-wind interactions and reduces eddy energy (Renault et al., 2016; Munday et al., 2021), BIOPERIANT12 EKE describes a surface SO with reasonable mesoscale eddy representation with which physical dynamics towards model improvement and BGC questions can be addressed.





### 3.1.3 Frontal structure

The SO fronts help describe the larger ACC structure, with their steep horizontal gradients and associated strong vertical motion, they demarcate regions of consistent water and nutrient properties, as well as regions of $CO_2$ in- and out-gassing: $CO_2$ in-gassing north of the Polar Front (PF) and out-gassing between the PF southwards to the marginal ice zone (Mongwe et al., 2018). Latitudinal variations in the frontal positions will thus reflect as local changes which affect heat and carbon fluxes and are therefore used as a SO evaluation metric by Russell et al. (2018), the positions of the SubAntarctic Front (SAF) and PF are chosen out of all the SO fronts, as representing the northern boundary and the central ACC, respectively. Their method uses a simplified subsurface temperature criteria consistent with (Orsi et al., 1995) which allows easy inter-model and model-observation comparisons: the SAF is defined by the 4 °C isotherm at 400 $m$ and the PF by the 2 °C isotherm in the upper 200 m. In Figure 3, we present the annual-mean position and standard deviation of the SAF and PF for BIOPERIANT12 interannual monthly-means, compared to those derived from WOA13 temperature (climatological months), the satellite-derived PF of Freeman and Lovenduski (2016), and Orsi et al. (1995), as used by Russell et al. (2018). In general, spatial meridional variabilities in the model frontal positions are consistent with the observation-derived fronts, but with a south bias by up to 3° in latitude, showing "pinching" of the fronts from topography such as Drake Passage, Campbell Plateau (170° E) and the Southwest (30° E) and Southeast (80° E) Indian Ridge, followed by diverging of the SAF and PF downstream. However, the difference in the model PF in the Indian Ocean stands out, highlighting the nature of averaging or the temporal under sampling of data. At the Kerguelen Plateau (∼75° E), the model follows either the northern or southern branch but the mean PF reflects the southern path (see Fig. S4) accompanied by very high standard deviation in contrast to data that show the mean path favours the northern branch over the northern plateau (Dong et al., 2006; Wang et al., 2016). As with the examples provided by Russell et al. (2018), the chosen models do not completely or consistently capture similar patterns to the observations, reflecting their individual model biases as well as their coarser resolution (CMIP5 models at ∼100 km). As shown in model EKE in Fig. 2, the mean mesoscale dynamics of the SO are captured by BIOPERIANT12, this includes the complex frontal dynamics described in observational studies such as branching, jets and front-eddy interaction (Freeman and Lovenduski, 2016; Chapman, 2017) which ultimately affects the calculated frontal positions particularly, a temperature-derived definition. The high variability in frontal dynamics of the mesoscale-resolving BIOPERIANT12, leads to model-observation differences, especially for regions such as the Kerguelen Plateau which is dominated by eddies as opposed to the meandering of the core jets (Shao et al., 2015); additionally, we expect a corresponding improved exchange of water and biogeochemical properties (Rosso et al., 2020). This complicates this metric for model-observation comparison but frontal position is still useful to delineate model regions for analysis.

### 3.1.4 Mixed Layer Depth

To evaluate the MLD of BIOPERIANT12, we chose in situ measurements given by the Argo floats database as our observational reference (Table 1). MLD from both sources were calculated from temperature and salinity values and using the de Boyer Montégut et al. (2004) density threshold of 0.03 kg m$^{-3}$ from a reference depth of 10 m, a method that has been shown to



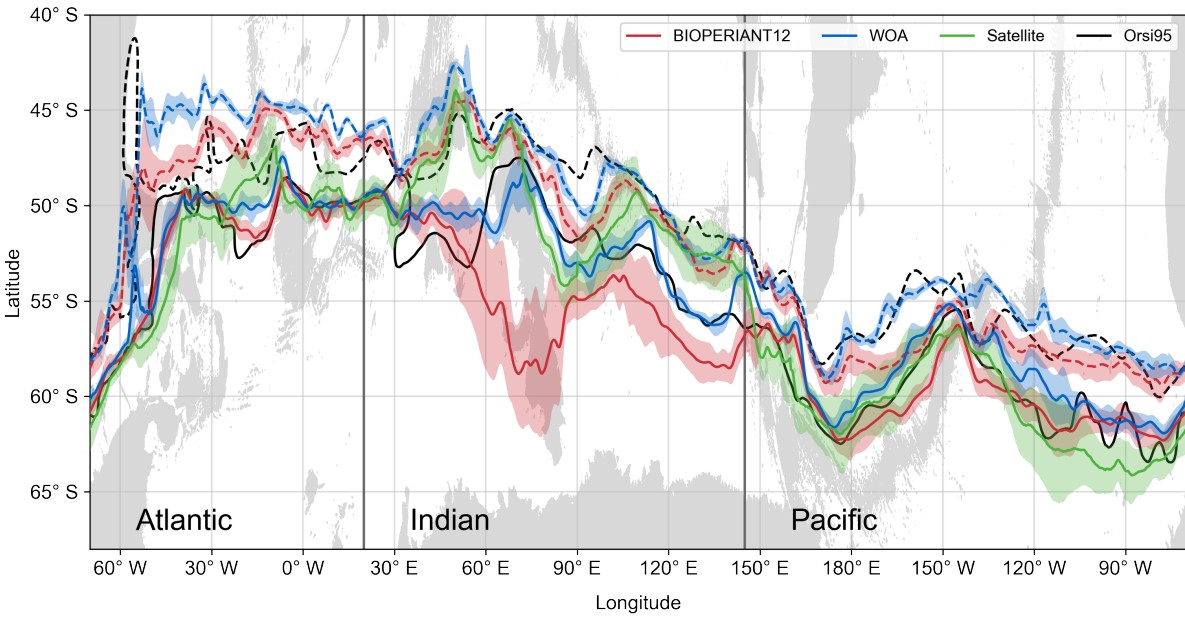

**Figure 3.** Annual mean latitudinal position of the SubAntarctic Front (dashed line) and Polar Front (solid line) in BIOPERIANT12 (2000–2009) compared to WOA13 and satellite Polar Front (2002–2009) from Freeman and Lovenduski (2016) and Orsi et al. (1995). Colour shaded regions are the standard deviation of the front using monthly-mean temperatures. Grey shaded regions indicate the land mask and bathymetry shallower than 3000 m (lighter grey).

be robust for Southern Ocean profiles (Dong et al., 2008; Treguier et al., 2023). While the Argo platform presents spatial and temporal gaps in their coverage (Fig. S5), the temperature and salinity profiles collected by the floats ultimately captures
210   the real-world variability and mixing features, in opposition to reanalysis products which introduce an unknown source of uncertainty depending on their reanalysis model. Model MLD was then sampled according to the Argo observational coverage. MLD observations were binned into a 1° x 1° regular grid and averaged to monthly intervals across the period 2001–2009. This was also done for model MLD which was then further subsampled to the same locations and months as observations. The seasonal-spatial patterns and amplitude are shown in Fig. 4.

215   Overall, the spatial distribution of the observational and BIOPERIANT12 MLDs agree well, for both the minimum and maximum MLD months, January and September, respectively (Fig. 4a–b, d–e). A comparison of the magnitude of the summer MLD (Fig. 4c) shows within the ACC, the BIOPERIANT12 MLD is too deep by around 50 m while it is too shallow by around 50 m in the north of the SAF. For winter MLDs, BIOPERIANT12 generates deeper MLDs relative to the observational estimates (by about 100 m; Fig. 4f) in the Pacific sector of the SO. Despite the magnitude difference, BIOPERIANT12 captures
220   the deep winter MLDs (exceeding 400 m) confined to the Pacific and Indian sectors showing that the model is responding to atmospheric forcing and forming deep waters in the expected regions. An important component of the MLD in the SO is the seasonal cycle (Fig. 4g). The winter deepening and summer shoaling of the simulated MLD agrees well with the seasonal



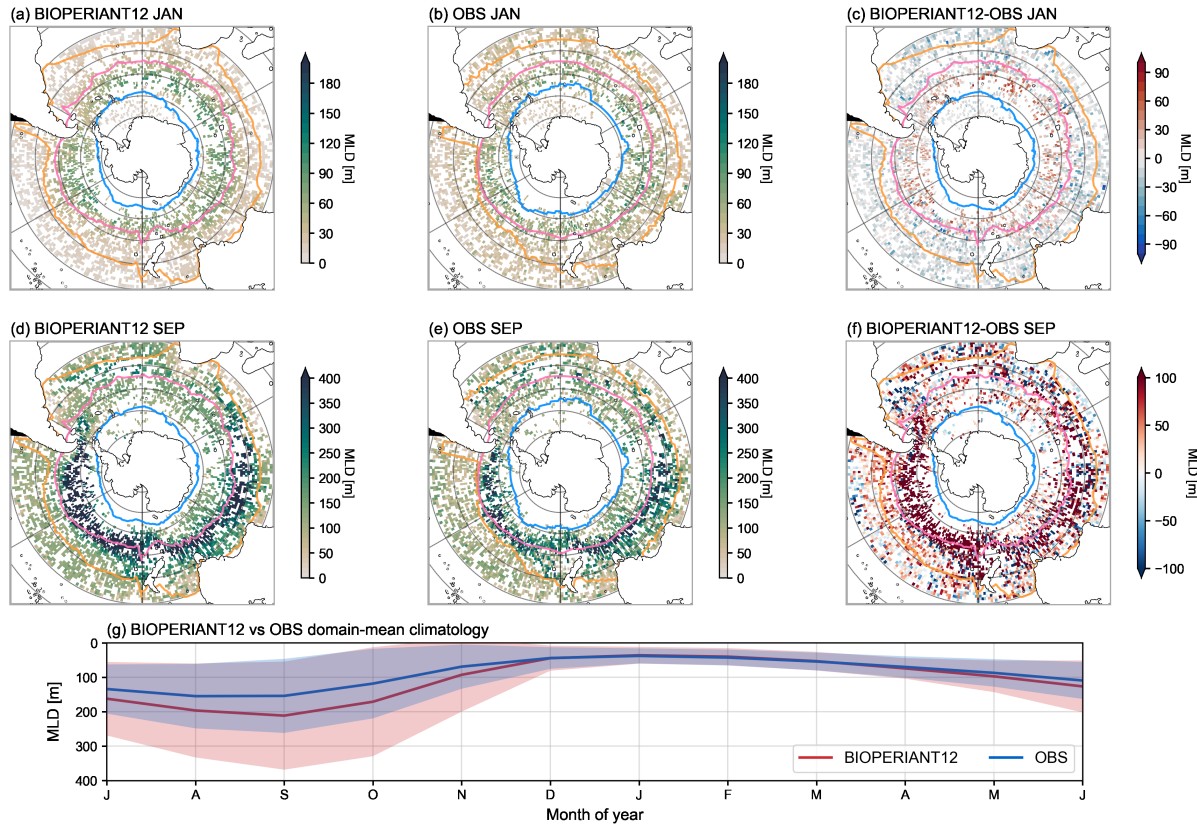

**Figure 4.** Seasonal comparisons of MLD from **(a, d)** BIOPERIANT12 averaged monthly and co-located to monthly averaged Argo MLDs in a $1° \times 1°$ regular grid, **(b, e)** Argo float observations, and **(c, f)** model–observation difference, for January and September climatologies respectively. Maps are overlaid with the northern climatological biome borders corresponding to model/data SO biomes. **(g)** Climatological seasonal cycle of monthly MLD domain-averaged over the entire SO.

cycle of the observations, with a monthly climatology Pearson correlation of r =0.97. The shallow limit of the simulated climatological MLD follows the observations well during DJFM, however the deep MLD limit departs from the observations around April, where standard deviations of the monthly mean MLDs show the BIOPERIANT12 MLD extending to over 100 m deeper in September (MLD maximum) than what the observations show. Treguier et al. (2023) showed that in OMIP models ranging from 1° down to 1/16° horizontal resolution, MLD is hard to constrain and increasing model resolution to include the mesoscale does not consistently improve MLD depth in the SO. Nevertheless, the seasonal cycle and variability of the BIOPERIANT12 MLD agrees well with the observations, despite over-deepening during winter.





### 3.1.5 Ocean Heat Content (OHC) and Temperature

BIOPERIANT12 temperature and OHC for the upper 400 m appear to compare well in seasonal distribution (Fig. S6). While no surface temperature correction has been applied to the model, it is relatively stable over the last 10 years of the model but model–data climatological differences between BIOPERIANT12 and WOA13 temperature and OHC show spatial variations in the mesoscale model fields (Fig. S7a, c). OHC in the model has a warm bias (domain mean around 13 x $10^9$ J m$^{-2}$, Fig. S7e), although SST shows a cool bias (Fig. S7b, d). While SO surface temperatures in models generally show warm biases (Beadling et al., 2020) and the opposite is shown for BIOPERIANT12 SST (Fig. 7f, domain-mean SST solid line), the deeper, domain-mean temperatures at 200 and 400 m are warmer than observations (Fig. S7f, dotted, and dashed lines) despite the low resolution, area-weighting of the dataset. The model–data differences are consistent over the run duration, the model is initialised with climatology and after spin-up shows no significant drift (Fig. S7g), the resultant stable upper ocean structure over the model analysis period suggests that the dynamics are also stable, possibly from increased mesoscale dynamics of the model which also influenced frontal definition above. Thus, we disregard the creation and influence of spurious features in the simulation that may propagate while altering the physical and biogeochemical fluxes, and any biogeochemical biases can be considered systematic or from model choices.

### 3.1.6 Sea ice

Sea ice plays an important role in setting the seasonal freshwater flux in the SO. Melting in the spring/summer and growth in autumn/winter sets the surface ocean salinity and thus has direct consequences for vertical and horizontal stratification dynamics (Giddy et al., 2021, 2023) and large-scale water mass transformation (Abernathey et al., 2016). For the model comparison, National Sea Ice Data Centre (NSIDC) sea ice observations mean monthly data for the period 2000–2009 were used (Table 1). Data shows sea ice grows northward from the Antarctic continent in the winter months, peaking with maximum extent in September and then melts towards the Antarctic continent to reach a minimum ice extent in February (Fig. 5). The spatial comparison for minimum/maximum climatological months (Fig. 5a, b) shows model maximum extent is comparable spatially to data but overestimates the sea ice growth by 1–2 million km$^2$, around 10 % of the winter sea ice maximum (Fig. 5c, d). Data shows minimum sea ice extent in February, while in the model minimum occurs over February and March with similar values. The timing of sea ice growth and melt by BIOPERIANT12 relative to the observations (Fig. 5c, d) indicates that the model responds well to seasonal heat fluxes with minimum and maximum extent agreeing with the observations, particularly in the winter. Although the model does not melt sufficiently in December (∼2 million km$^2$ relative to ∼5 million km$^2$ from the observation), it compensates for this by not melting the remainder of the necessary sea ice in February (∼5 million km$^2$ relative to ∼2 million km$^2$ from the observations) before both the model and observations show sea ice growth from March (Fig. 5d; S8b). Overall, the sea ice in BIOPERIANT12 provides a stable temporal view of the seasonal cycle, with more interannual variability seen in the summer (Fig. S8b).

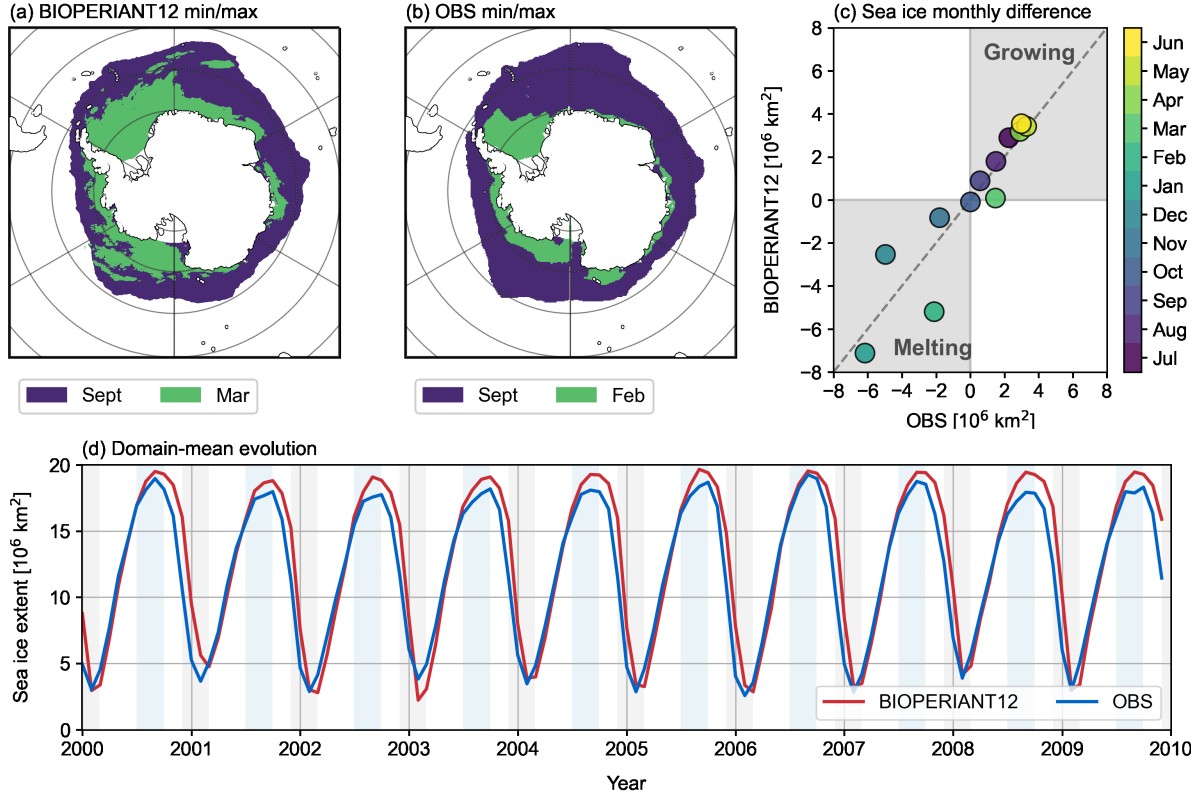

**Figure 5.** Climatological-mean sea ice extent of the maximum (blue) and minimum (green) months for **(a)** BIOPERIANT12 and **(b)** NSIDC observations. **(c)** Climatological monthly-mean sea ice extent difference indicating growth and melt for the BIOPERIANT12 simulation vs NSIDC observations. **(d)**Timeseries of monthly-mean sea ice extent from BIOPERIANT12 model (red) and NSIDC observations (blue).

## 3.2 Biomes

### 3.2.1 Biome definition

To evaluate the biogeochemical and carbon fields, we chose to use the biome classification method (Fay and McKinley, 2014), over static geographic definitions or that from ocean physics (e.g. by fronts, a dynamical field), thus combining both large scale physical and biogeochemical characteristics to delineate regions of biogeochemical similarity (biomes). BIOPERIANT12 mean biomes (Fig. 6a) were calculated using the model's climatological sea surface temperature (SST), Chlorophyll-a, MLD and sea ice fraction fields. The Fay and McKinley (2014) biome product based on 1998–2010 climatology (Fig. 6b) will be used to aggregate observational data. We thus compare similar, biophysically consistent SO regions between the simulated variables and the observations for the following biomes: the ice biome (SO-ICE), the subpolar seasonally stratified biome (SO-SPSS), and the subtropical seasonally stratified biome (SO-STSS). Other southern hemisphere biomes were not considered, BIOPERIANT12 does not fully capture the subtropical mean state (SP-STPS, SA-STPS and IND-STPS) which results in about



60 % less area coverage than the corresponding observed biomes and slightly overestimates the SO-STSS, SO-SPSS (Fig. 6c). Thus, the biome criteria may suggest some of the model–observation gaps (Fig. S9, S10) as well as address the effect of mesoscale features that are resolved in the higher resolution model compared to the gridded data used in the Fay and McKinley
(2014) biome product and is discussed in the Supplementary.

### 3.2.2 Biome characteristics

To characterise the seasonal cycle of the model, we present the surface mean seasonal cycle of selected variables for each SO biome (Fig. 7) and biome-mean metrics in Table 3 (Table S1 for 200 m). To contextualise the upper water column of the biome, particularly the influence on surface $pCO_2$ in highly seasonally stratified areas, vertical profiles per biome are provided
in Fig. S11. In Table 3, the biome-mean seasonal cycle for model and observations (as shown in Fig. 7) are given by mean and amplitude (difference between the minimum and maximum values). To compare the model's seasonal cycle and that of the observational data, we used: the correlation coefficient between model and data (R), ratio of standard deviation (RSD), model Reliability Index (RI), and Seasonal Cycle Reproducibility (SCR). The RSD shows how the model variability compares to that expected by the observed dataset, a value of one suggests they compare well. We highlight certain regions in the model
which are inconsistent with data by regions where the model variability exceeded that of observations by 50 % and where model variability was less than 50 % of the expected variability. The model RI or geometric root-mean square error (Leggett and Williams, 1981; Doney et al., 2009), gives the model–data bias that is normalised to the data and used for data that are log-normal distributed; a value of one indicates perfect agreement, while two indicates that the error is of similar magnitude to the data. The metric SCR is defined in Thomalla et al. (2011) as the correlation coefficient of the interannual varying time
series at its full temporal resolution against its climatological seasonal cycle (Fig. S12 for further explanation), i.e. how well the mean climatological seasonal cycle represents the evolution of a property over each year. SCR higher than 0.85 defines a region of high SCR (e.g. high seasonality), 0.65 to 0.85 medium SCR and below 0.65, low SCR (e.g. low seasonality). The SCR for observational products were only calculated if they had an adequate temporal resolution (higher than monthly).

High resolution observations show that the seasonal response of BGC to physical forcing in the SO are regionally dependent.
Model–data agreement results from a combination of factors: the available/applied observational dataset (listed in Table 1); which carries uncertainty due to poor sampling/spatiotemporal resolution and are oftentimes interpolated into gridded products and only available as climatologies; the application of biome definitions over regions of high variability (EKE) which is not used for biome definition but is shown to improve $pCO_2$ bias and root mean square error (Gregor et al., 2019); and the effect of area-weighted, domain averaging over large regions/biomes. In contrast to regions dominated by intraseasonal variability
where observations are difficult to constrain (Monteiro et al., 2015), the seasonal cycle metrics are expected to agree with the data in regions where the response of the model is also seasonally driven with high SCR.

In general, the metrics (Table 3, Fig. 7) show the model seasonal cycle agrees with that of the data, for example, the temperature seasonal cycle per biome is well represented, however, a reasonable comparison with the data is expected since temperature is a criterion used in the biome definition. We, therefore, focus on the model-observation disagreements which
highlight regions of interest. For the dynamics, there is a poor correlation of salinity in the SO-STSS region (R=0.62), this

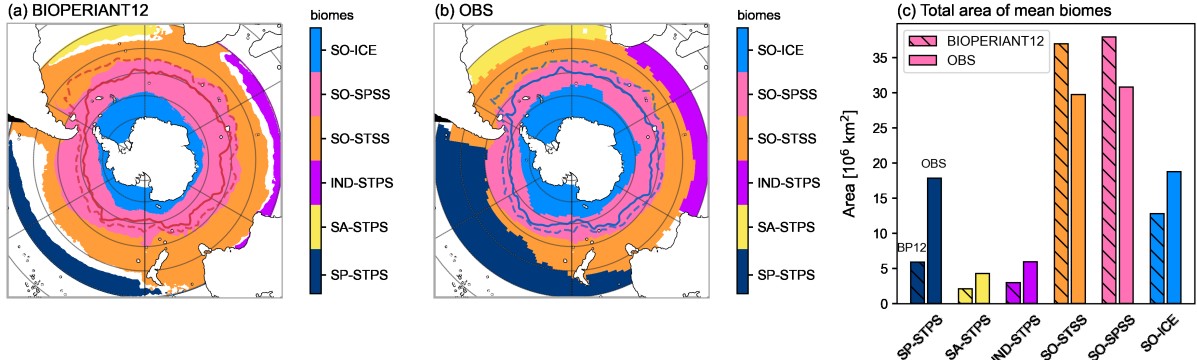

**Figure 6.** SO mean biome boundaries for **(a)** BIOPERIANT12 using the biome criteria definitions from Fay and McKinley (2014), **(b)** the observation-based mean biome dataset of Fay and McKinley (2014) south of 30° S (see Supplementary). **(c)** Total area per biome for BIOPERIANT12 (hatched bars, titled BP12) and for the dataset (plain bars, titled OBS). In the SO (below 30° S), six biomes are identified: the ice biome (SO-ICE), the subpolar seasonally stratified biome (SO-SPSS), the subtropical seasonally stratified biome (SO-STSS) and the Indian, South Atlantic, and South Pacific subtropical permanently stratified biome (IND-STPS, SA-STPS and SP-STPS respectively).

could be associated with the high mesoscale spatial and temporal variability (low seasonality of 0.12) in the model compared with data. While in the SO-ICE biome, high variability in salinity (RSD=1.55), MLD (RSD =1.56) as well as dissolved $O_2$ (RSD=1.66), point at the influential role of sea ice and freshwater dynamics in the model region and suggest improvement or further investigation. For the BGC, poor model reliability for carbon, silicate, and chlorophyll as well as deviations in nutrients will be addressed in the proceeding sections.





**Table 3.** Seasonal cycle surface climatology per biome: BIOPERIANT12 (BP12) vs observations (OBS): area-weighted Mean and Amplitude (difference between maximum and minimum) for the respective data sources; SCR seasonal cycle reproducibility (the correlation of the interannual varying timeseries against its climatological seasonal cycle) where temporal resolution allows; and for comparison Correlation (correlation coefficient), RSD (ratio of BP12:OBS standard deviation), Model Reliability Index (log-transform error).

| Variable | Biome | SO-ICE BP12 | SO-ICE OBS | SO-SPSS BP12 | SO-SPSS OBS | SO-STSS BP12 | SO-STSS OBS |
|---|---|---|---|---|---|---|---|
| Temperature $[°C]$ | Mean | -1.49 | -1.00 | 3.03 | 3.76 | 13.45 | 12.75 |
| | Amplitude | 1.16 | 1.82 | 2.36 | 2.74 | 3.90 | 4.25 |
| | SCR | 0.63 | | **0.71** | | 0.54 | |
| | Correlation | | 0.94 | | 0.98 | | 0.99 |
| | RSD | 0.66 | | 0.90 | | 0.93 | |
| | Model RI | **2.32** | | 1.26 | | 1.06 | |
| MLD $[m]$ | Mean | 66.72 | 57.02 | 109.79 | 109.08 | 86.20 | 100.53 |
| | Amplitude | 86.87 | 54.57 | 143.26 | 118.04 | 153.40 | 158.03 |
| | SCR | **0.76** | | **0.74** | | **0.75** | |
| | Correlation | | 0.86 | | 1.00 | | 1.00 |
| | RSD | **1.56** | | 1.19 | | 0.95 | |
| | Model RI | 1.41 | | 1.11 | | 1.31 | |
| pCO$_2$ $[\mu atm]$ | Mean | 345.25 | 367.37 | 363.98 | 365.15 | 362.96 | 345.27 |
| | Amplitude | 49.30 | 47.90 | 10.22 | 17.10 | 13.08 | 11.85 |
| | SCR | **0.79** | **0.79** | 0.59 | **0.80** | 0.51 | **0.83** |
| | Correlation | | 0.89 | | **-0.45** | | **-0.59** |
| | RSD | 0.91 | | 0.65 | | 1.06 | |
| | Model RI | 1.07 | | 1.02 | | 1.06 | |
| DIC $[\mu mol kg^{-1}]$ | Mean | 2181.93 | 2148.95 | 2151.61 | 2123.52 | 2087.99 | 2056.03 |
| | Amplitude | 62.96 | 0.00 | 15.50 | 0.00 | 23.00 | 0.00 |
| | SCR | 0.53 | | 0.44 | | 0.47 | |
| | Correlation | | | | **0.00** | | **0.00** |
| | RSD | | | | | | |
| | Model RI | 1.02 | | 1.01 | | 1.02 | |
| Nitrate $[mmol l^{-1}]$ | Mean | 26.02 | 25.73 | 23.04 | 21.75 | 10.52 | 6.83 |
| | Amplitude | 3.92 | 3.69 | 1.91 | 2.73 | 3.77 | 3.52 |
| | SCR | 0.58 | | 0.46 | | 0.61 | |
| | Correlation | | 0.95 | | 0.95 | | 0.97 |
| | RSD | 1.28 | | 0.73 | | 1.24 | |
| | Model RI | 1.02 | | 1.06 | | **1.55** | |
| Silicate $[mmol l^{-1}]$ | Mean | 48.47 | 53.42 | 26.46 | 15.85 | 5.81 | 3.37 |
| | Amplitude | 13.96 | 12.44 | 5.47 | 11.30 | 2.70 | 2.70 |
| | SCR | 0.64 | | 0.49 | | 0.62 | |
| | Correlation | | 0.85 | | 0.96 | | 0.95 |
| | RSD | 1.18 | | **0.49** | | 1.05 | |
| | Model RI | 1.12 | | **1.79** | | **1.79** | |

| Variable | Biome | SO-ICE BP12 | SO-ICE OBS | SO-SPSS BP12 | SO-SPSS OBS | SO-STSS BP12 | SO-STSS OBS |
|---|---|---|---|---|---|---|---|
| Salinity | Mean | 34.03 | 33.96 | 34.01 | 33.93 | 34.83 | 34.71 |
| | Amplitude | 0.70 | 0.43 | 0.08 | 0.08 | 0.01 | 0.06 |
| | SCR | 0.35 | | 0.20 | | 0.12 | |
| | Correlation | | 0.99 | | 0.84 | | 0.62 |
| | RSD | **1.55** | | 1.07 | | **0.14** | |
| | Model RI | 1.00 | | 1.00 | | 1.00 | |
| Tot. chl-a $[mg m^{-3}]$ | Mean | 0.58 | 0.28 | 0.51 | 0.24 | 0.61 | 0.35 |
| | Amplitude | 1.69 | 0.70 | 0.88 | 0.23 | 0.75 | 0.21 |
| | SCR | **0.88** | 0.36 | **0.85** | 0.36 | **0.68** | 0.21 |
| | Correlation | | 0.93 | | 0.86 | | 0.91 |
| | RSD | 2.37 | | **4.22** | | **3.64** | |
| | Model RI | **1.86** | | **2.10** | | **1.69** | |
| FCO$_2$ $[mol m^{-2} yr^{-1}]$ | Mean | -0.82 | 0.15 | -1.12 | -0.07 | -0.77 | -1.81 |
| | Amplitude | 2.17 | 1.80 | 1.17 | 1.48 | 1.54 | 0.45 |
| | SCR | 0.39 | **0.77** | 0.31 | **0.80** | 0.28 | **0.76** |
| | Correlation | | 0.89 | | **-0.78** | | **-0.23** |
| | RSD | 1.30 | | 0.80 | | **3.53** | |
| | Model RI | **3.87** | | **2.59** | | **6.37** | |
| Tot. Alkalinity $[\mu mol kg^{-1}]$ | Mean | 2303.80 | 2295.51 | 2295.77 | 2281.70 | 2317.60 | 2300.31 |
| | Amplitude | 43.62 | 0.00 | 3.67 | 0.00 | 4.02 | 0.00 |
| | SCR | 0.50 | | 0.29 | | 0.21 | |
| | Correlation | | **-0.00** | | | | |
| | RSD | | | | | | |
| | Model RI | 1.01 | | 1.01 | | 1.01 | |
| Phosphate $[mmol l^{-1}]$ | Mean | 1.85 | 1.81 | 1.63 | 1.56 | 0.80 | 0.68 |
| | Amplitude | 0.22 | 0.26 | 0.10 | 0.32 | 0.21 | 0.33 |
| | SCR | 0.56 | | 0.42 | | 0.55 | |
| | Correlation | | 0.99 | | 0.90 | | 0.92 |
| | RSD | 0.86 | | **0.36** | | 0.67 | |
| | Model RI | 1.03 | | 1.07 | | 1.22 | |
| Diss. O$_2$ $[\mu mol l^{-1}]$ | Mean | 357.13 | 347.78 | 332.14 | 326.98 | 264.86 | 274.04 |
| | Amplitude | 29.65 | 21.34 | 13.77 | 14.52 | 17.08 | 16.90 |
| | SCR | 0.53 | | 0.45 | | 0.47 | |
| | Correlation | | 0.86 | | 0.94 | | 0.99 |
| | RSD | **1.66** | | 0.91 | | 0.95 | |
| | Model RI | 1.03 | | 1.02 | | 1.03 | |

[a] GLODAPv2 observations are an annual mean product.

**RSD**: SD of BP12 greater (less) than that of OBS by factor 1.5 (0.5). **Correlation**: BP12-OBS correlation coefficient less than 0.5.

**SCR**: BP12 greater than 0.65 (medium to high). **Model RI**: greater 1.5 (medium to high).





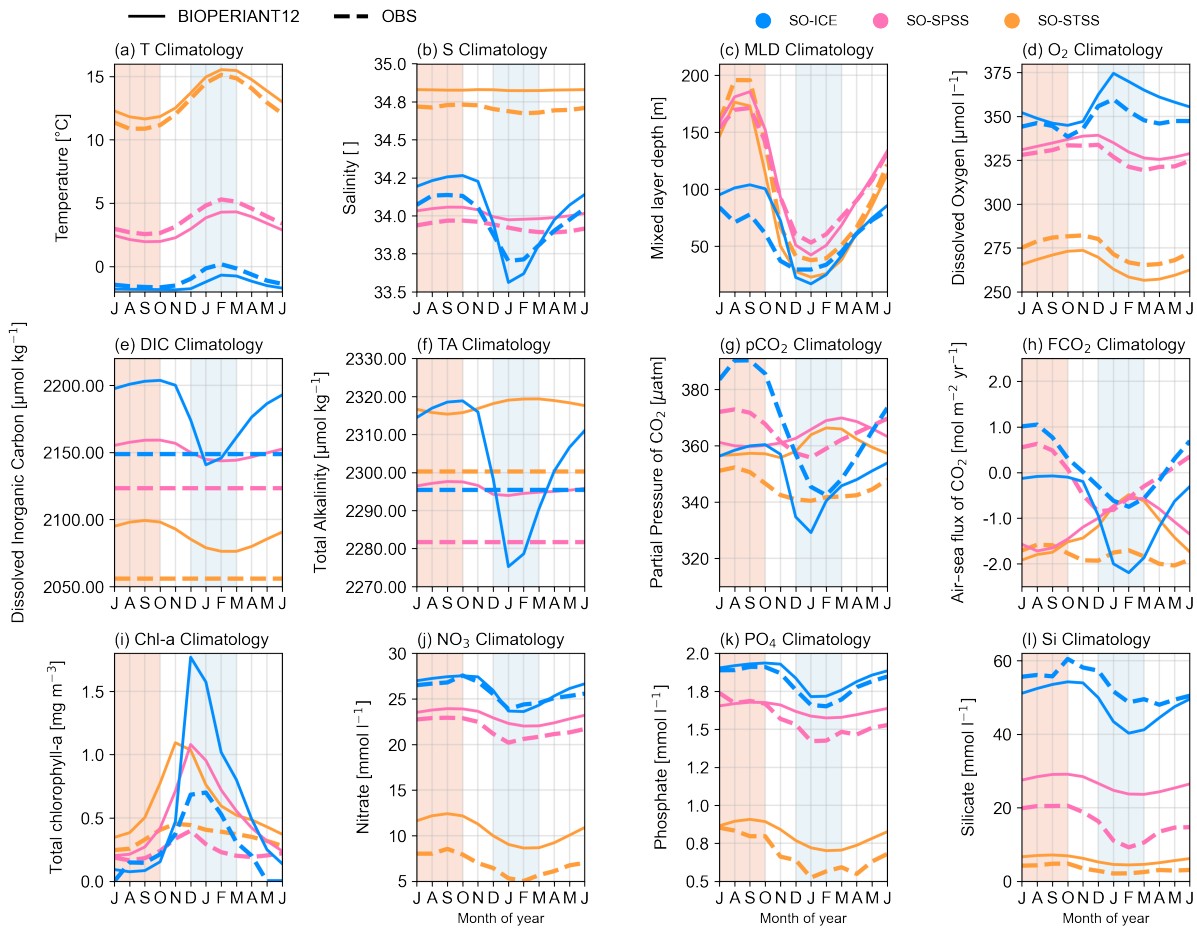

**Figure 7.** BIOPERIANT12 model (solid line) versus Observations (dashed line) surface seasonal cycle (2000–2009 climatology) spatially averaged per biome for selected variables. Biomes include SO-ICE (blue), SO-SPSS (pink) and SO-STSS (orange) corresponding to map in Fig. 6.





## 3.3 Carbon

Recent studies have highlighted the importance of resolving intra-seasonal to seasonal modes of variability for both anthropogenic and natural carbon fluxes in order to reduce the uncertainties of the mean annual fluxes and strengthen model projections (DeVries et al., 2023; Rustogi et al., 2023). Using SCR, as a metric of variability, a comparison of the SCR of pCO2

in BIOPERIANT12 against the monthly, data product CSIR-ML6 (Fig. 8) shows that model $pCO_2$ for the SO-STSS and SO-SPSS biomes are dominated by large regions of interannual variability (SCR=0.59, 0.51, respectively) aligned with regions of high EKE (Fig. 2) which are not captured by the monthly CSIR-ML6 product (Fig. 8c). CSIR-ML6, instead, displays high seasonality in all three SO biomes with SCR 0.80, 0.80, 0.79 in the SO-STSS, SO-SPSS, SO-ICE biomes, respectively (Table 3), i.e. driven by seasonal surface fluxes and ice (although the monthly temporal resolution may contribute to this).

This results in dissimilar $pCO_2$ seasonal cycles (Fig. 7g) highlighted by weak correlations in the SO-STSS and SO-SPSS biomes (R=-0.59, -0.45, respectively) whilst showing good model–data agreement in the SO-ICE biome; a region that is also strongly seasonally driven in the model (SCR=0.79, for both model and data) and with coherent model–data phasing (R=0.89), although minima are one month out of phase. Despite biome mean differences in the model climatology, we calculate a small model–data bias of $pCO_2$ for all 3 biomes indicated by model RI between 1.02 to 1.07, which suggests that BIOPERIANT12

is in agreement with the data product CSIR-ML6; also shown in the Probability Density Functions (PDFs) for $pCO_2$ for the SO domain mean (Fig. S13a). This emphasises the importance of mesoscale-resolving model resolution in the SO, in capturing the variability of $CO_2$.

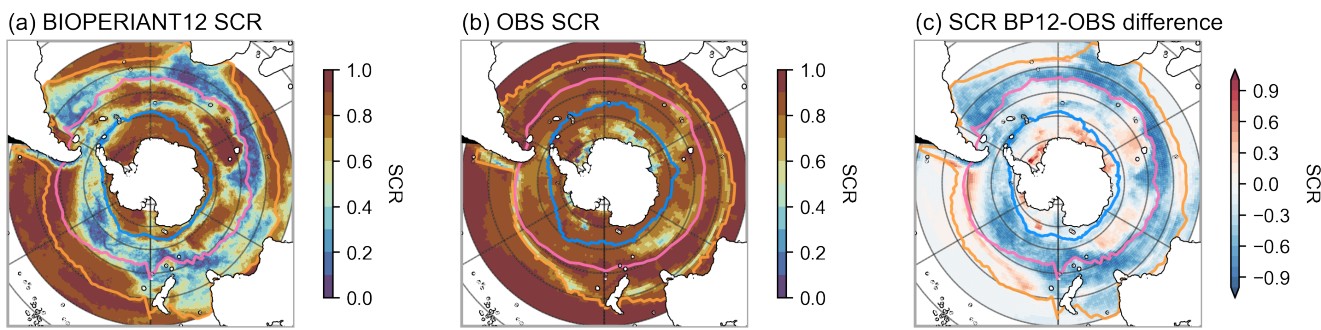

**Figure 8.** Seasonal cycle reproducibility of $pCO_2$ for **(a)** BIOPERIANT12 , **(b)** CSIR-ML6 observation based reconstruction, and **(c)** model–data difference. Maps are overlaid with the northern climatological biome borders corresponding to model/data SO biomes.

The spatial distribution of $pCO_2$ shown in Fig. 9 (a–f) reflects the seasonality differences between model and data, showing the overestimation of model versus data $pCO_2$ in the SO-STSS and underestimation in the SO-ICE biomes. Differences in sea-

sonal cycle phasing in the SO-SPSS and SO-STSS biomes are clearly seen in the interannual time series (Fig. 9i, k) with model peaks in summer compared to winter peaks in the data. The PDFs for $pCO_2$ (Fig. 9h, j, l) show a general wider distribution (higher standard deviation) in the model biomes and not the distinct inter-biome differences shown in the data, such as the narrow $pCO_2$ distribution in the SO-SPSS (Fig. 9j).





The seasonal cycle differences of $pCO_2$ (and $FCO_2$) in the model SO-SPSS and SO-STSS biomes relative to these ob-
servations may be representative of different mechanisms. The $pCO_2$ seasonal maximum in summer (JFM) is in phase with
temperature maximum (Fig. 7a) but out of phase with the chlorophyll peak 2 months earlier (OND) (Fig. 7i, 13) and associated
nutrient decrease ($NO_3$ and Si, Fig. 7j, l). This is consistent with the model's $pCO_2$ seasonal cycle in the SO-SPSS and SO-
STSS biomes being primarily regulated by the thermal component of $pCO_2$ (Mongwe et al., 2016). On the other hand, there is
better model–data phasing of the seasonal cycles of $pCO_2$ and $FCO_2$ in the SO-ICE biome (Fig. 7g, h). Additionally, temper-
ature, chlorophyll, and nutrient model-data seasonal cycles are comparable in phase (Fig. 7i–l). However, the amplitude of the
BIOPERIANT12 seasonal chlorophyll maximum is double that of the observed estimate (due to regions of high chlorophyll,
Fig. 11) which may result in a larger contribution of the biological components driving $pCO_2$ as compared to the thermally
driven ice free regions.

It is noted that observational products, particularly underway $pCO_2$ observations used in reconstructions, have significant
seasonal biases themselves, as they are based on limited observations during winter. Thus, the magnitude and direction of
the observed seasonal cycle of $pCO_2$ and $FCO_2$ in the SO is still under investigation (Gray et al., 2018; Landschützer et al.,
2018; Gregor et al., 2019; Gruber et al., 2019; Bushinsky et al., 2019; Mackay and Watson, 2021). The seasonal biases in
these products may also manifest as artificial variability on longer time scales (Hauck et al., 2023). The poor representation of
biological processes of the SO-SPSS and SO-STSS in the model contributing to differences in the $pCO_2$ (and $FCO_2$) seasonal
cycles is discussed in the Supplementary. We propose that a weaker seasonal vertical flux in DIC as shown by the smooth upper
ocean DIC gradient compared to the dataset (Fig. S11) leads to a weaker DIC entertainment potential during the seasons of
enhanced vertical mixing, and hence a diminished seasonal DIC variability (Mongwe et al., 2016).

## 3.4 Biogeochemistry

### 3.4.1 Dissolved Iron

Dissolved iron (dFe) limits phytoplankton growth across the surface of the SO, impacting the functioning of marine ecosystems
and thus the carbon cycle. It is thus imperative that models adequately represent the spatial and seasonal distribution of dFe.
In the SO, dFe is notoriously undersampled, it occurs at small (nanomolar, n M) concentrations and its complex chemistry
makes it difficult to observe. Here, we compare BIOPERIANT12 with a compilation of dFe observations collated by Tagli-
abue et al. (2012) over 2000–2009 (Fig. 10). BIOPERIANT12 simulates the observed spatial distribution of upper ocean dFe
concentrations, with higher concentrations > 0.4 n M close to coastal boundaries, downstream and around subAntarctic islands
(particularly evident for Kerguelen), and in the vicinity of the Agulhas Retroflection (Fig. 10a, c). Far from the influence of
land masses, the simulated dFe concentrations are lower < 0.3 n M, as similarly seen in the observations (Fig. 10b). In general,
the simulated range of upper dFe is on the lower end of the range of the observations; it has been noted that PISCES tends to
exaggerate the low dFe concentrations in the open ocean of the SO (Aumont et al., 2015).

The shape of the vertical profile of dFe is important as it plays a fundamental role in how much dFe is available to be
supplied to the surface i.e. via deep winter convective mixing and by mesoscale eddies (Tagliabue et al., 2014; Nicholson





**Figure 9.** Seasonal comparison of pCO$_2$ for **(a, d)** BIOPERIANT12, and **(b, e)** CSIR-ML6 observation based dataset, and **(c, f)** model–data difference, for January and September monthly climatology, respectively. Maps are overlaid with the northern climatological biome borders corresponding to model/data SO biomes. Evolution of area-weighted pCO$_2$ for model versus data and correspnding PDF for **(g, h)** SO-ICE, **(h, i)** SO-SPSS, and **(k, l)** SO-STSS biomes.

et al., 2019). BIOPERIANT12 is able to simulate the mean observed characteristics of a general dFe profile (Fig. 10e) with low concentrations between 0-100 m due to biological consumption and increasing DFe concentrations with depth due to remineralization of sinking organic material. The simulated mean range dFe compares relatively well with mean observations over the upper 500 m. Below this, the simulated mean range is lower. The lack of measurements during key seasonal transitions






particularly over winter (Fig. 10f) make comparisons of the simulated seasonal distribution of surface dFe difficult. However, during austral summer (DJF), dFe is expected to be low due to biological consumption from spring to summer and during winter (JJA), dFe is expected to be higher due to deep convective mixing entraining subsurface dFe to the surface (Tagliabue et al., 2012) These seasonal differences are simulated by BIOPERIANT12 (Fig. 10f) and are particularly evidenced by the

summer and winter snapshots in Fig. 10c and d, respectively.

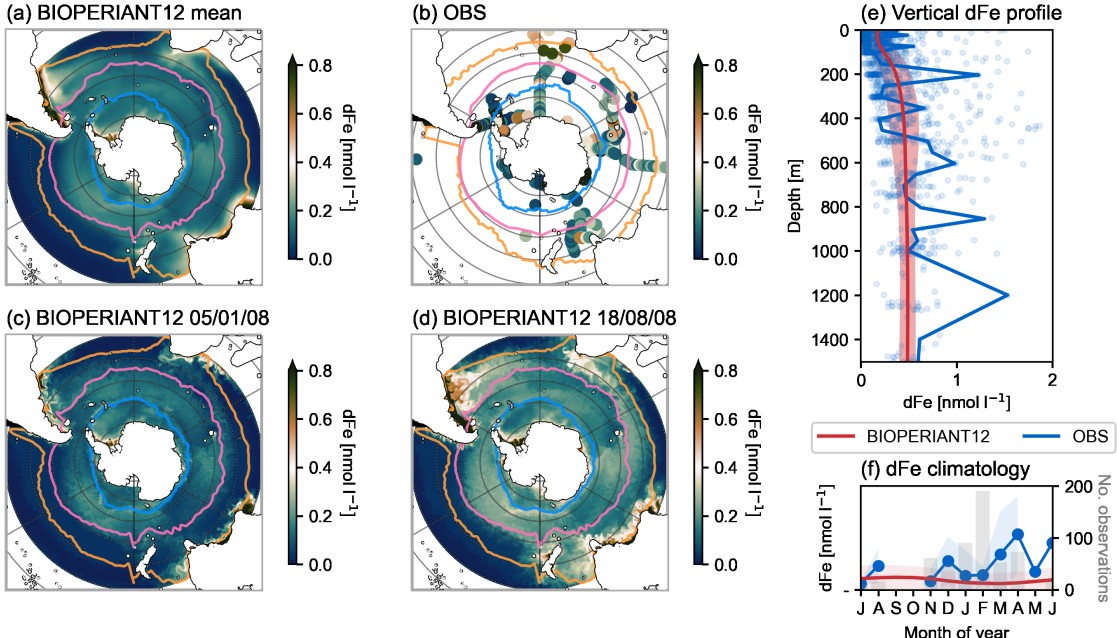

**Figure 10.** Surface (0-50 m) dissolved iron (dFe) concentration for **(a)** BIOPERIANT12 climatological annual mean **(b)** observations obtained during years 2000-2009 (Tagliabue et al., 2012). Seasonal snapshots of BIOPERIANT12 dFe for **(c)** summer (5 January 2008) and **(d)** winter (18 August 2008). **(e)** Annual mean vertical distribution of dFe for BIOPERIANT12 without area-weighting (red line = mean, red shading = spatial standard deviation) and observations (blue). **(f)** Model versus observation monthly mean climatology (line) and standard deviation (shading). The number of observations in each month is shown by grey bars.

### 3.4.2  Nutrients

Model RI values in Table 3 show that the model is able to reasonably reproduce the seasonal climatology of $NO_3$ and $PO_4$ with relatively comparable seasonal cycles (Fig. 7), particularly for the more southern biomes (e.g. RI values between 1.02 and 1.07 for both $NO_3$ and $PO_4$ in the SO-ICE and SO-SPSS biomes). However, simulated surface silicate is less well represented by the

model (SO–SPSS and SO–STSS both RI=1.79): in the SO–SPSS, model silicate means differ compared to data (26.46 vs. 15.85 mmol l⁻¹), considering their amplitudes (5.47 vs. 11.3 mmol l⁻¹) and variability that is half that of the data (model RSD=0.49). These differences may result from the representation of simulated silicate due to uncertainty in laboratory experiments as noted





in Aumont et al. (2015). This may affect the concentration of diatoms in the model. $PO_4$ variability in the model is also low, 0.36 times that expected of data however, with comparable means. It is important to note that much of the observational data is collected during the productive season (austral summer) in the SO, and thus surface values observed may be biassed towards lower values (Fig. 7, S16). However, deeper in the water column at 200 m (Table S1, Fig. S17), RIs for the 3 aforementioned nutrients reflect a good agreement between the model and observations at this level (between 1.00 and 1.30). Model nutrients at this depth are stable, standard deviations smaller than from the data climatology as in the vertical profiles (Fig. S11). Over the evolution of the model (Fig. S16 and S17), there is a slight decreasing trend in nutrients, particularly noticeable in the SO-SPSS at the surface and 200 m and should be taken into account in process studies.

### 3.4.3  Surface chlorophyll

BIOPERIANT12 represents the main spatial surface patterns of regions of higher and lower surface chlorophyll as compared to OC-CCIv6 satellite-derived chlorophyll concentrations (Fig. 11). For example, it simulates the higher surface chlorophyll close to continental margins and in the region of ocean fronts such as in the SAZ (not shown), while low chlorophyll is simulated in more oligotrophic regions such as the SO Pacific. However, the spatial extent of enhanced chlorophyll regions associated with shallow topography are larger in the model. The incorporation of these regions within the biome definitions contribute to the magnitude of the surface chlorophyll which is overestimated in BIOPERIANT12, with variability over double that of the observations in all biomes (RSD greater than 2 in Table 3).

SCR (Fig. 12) suggests that the drivers of the differences are from dynamics as model chlorophyll is seasonally driven while the data product suggests that drivers are on the intraseasonal time-scale. Despite this, the seasonal cycle for the SO biomes correlates well with data. A comparison of the simulated biome-mean seasonal cycle against observations of OC-CCIv6 chlorophyll for each biome (Fig. 13a–c, S18) shows good agreement of the timing of the bloom maximum in December (date difference within 14 days). However, the general characteristics, such as initiation and termination date defined using the biomass threshold method of 5 % (Ryan-Keogh et al., 2023) only compare well for the SO-STSS region, both initiating in August and terminating in May. For the SO-SPSS biome, both the model and observation blooms start in September and while OC-CCIv6 blooms terminate in March, the model sustains the bloom a further 93 days into June. In contrast, the SO-ICE biome bloom for both sources terminate in May, while the simulated bloom starts 3 months after observations (October vs. August).



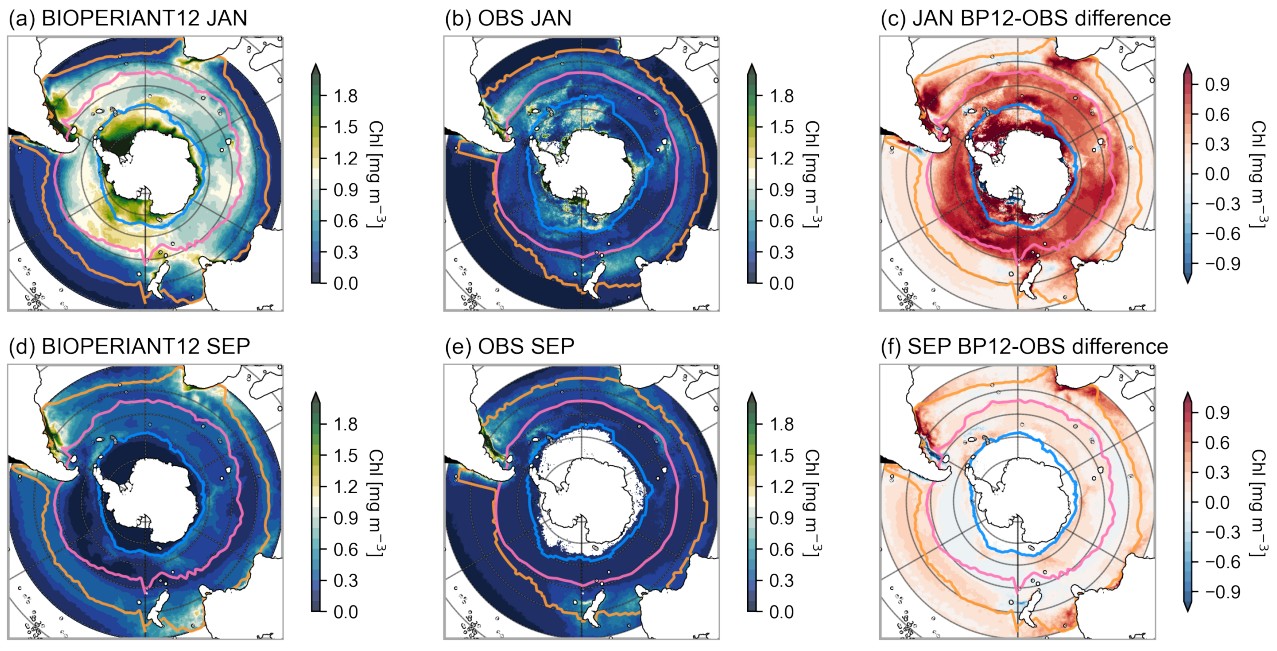

**Figure 11.** Climatological mean total chlorophyll concentrations for **(a, b)** BIOPERIANT12 vs **(d, e)** OC-CCI observation-based product for January and September. SCR of surface chlorophyll for **(c)**BIOPERIANT12 and **(f)** observations. Model versus observation seasonal cycle of surface chlorophyll for **(g)** SO-ICE, **(h)** SO-SPSS, and **(i)** SO-STSS biomes.

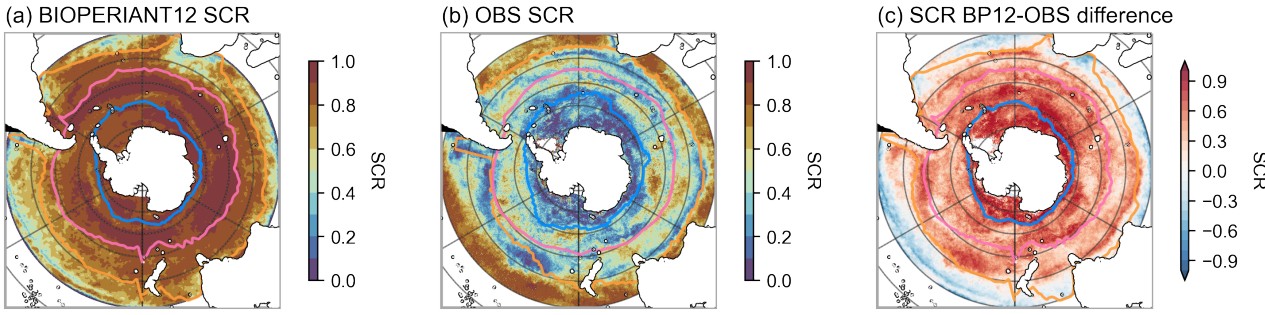

**Figure 12.** Seasonal cycle reproducibility (SCR) of surface chlorophyll concentration for (a) BIOPERIANT12, (b) OC-CCIv6 observation-based product, and (c) model-observation SCR bias.



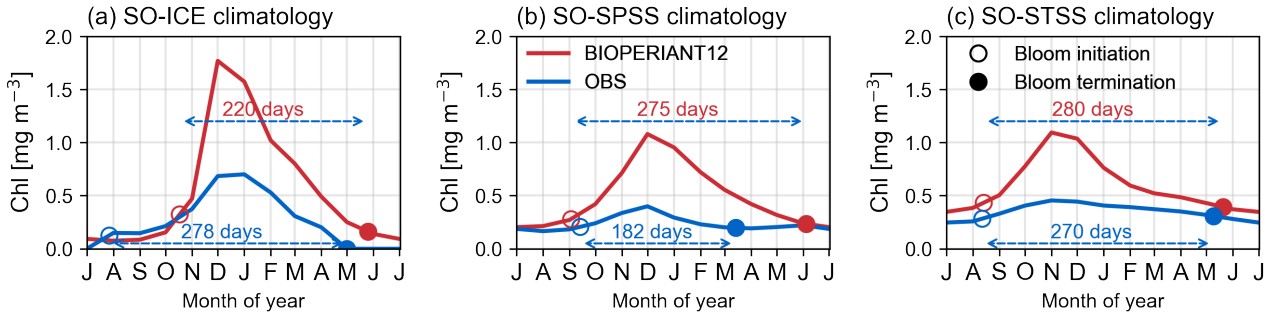

**Figure 13.** BIOPERIANT12 model versus OC-CCI observation climatological bloom characteristics overlaid on the seasonal cycle of surface chlorophyll for (a) SO-ICE, (b) SO-SPSS, and (c) SO-STSS biomes.





## 4   Conclusions

The complex dynamics of the SO has been scrutinised in many papers, particularly with its role in the climate system and influence on carbon projections by ESMs. While inter-model comparisons may show the weaknesses and strengths, the models themselves are also complex systems defined by multiple interacting parts defined by many assumptions, making isolating biases challenging. BIOPERIANT12, on the other hand, while composed of the major components of ocean, ice and biogeochemistry, provides a simpler, stable reference model with a reasonable mean state with which we can address process

questions, diagnose model biases, and build future model experiments. Despite the eddying resolution, the simulated SO seasonal cycle still needs improvement. Model-observation differences, not addressed in this evaluation paper, can lead to further paths of enquiry.

The importance of the mesoscale on biology in the SO requires particular attention with the local environment of the phytoplankton modified through both biological and physical mechanisms, for example by altering gradients and adjusting access

to light and nutrients which affect growth and hence the biological carbon pump. Large biomass anomalies in both spatial and seasonally variability are reported in Rohr et al. (2020). In BIOPERIANT12, chlorophyll variability from two phytoplankton classes (nanophytoplankton and diatoms), averaged over biomes, exceeded observational values by more than double. In the SO, where biological processes play an important role in carbon exchange, the simulated processes which give rise to these high values need to be explored. Despite the magnitude differences, the temporal variability should also be addressed, the

SCR shows that the simulated chlorophyll response is driven by strong seasonal factors, when a higher-temporal response is expected as displayed by the OC-CCIv6 data.

BIOPERIANT12 provides a stable, coherent 3–dimensional evolving dataset at relatively high resolution to test sampling strategies and biases, such as investigating the uncertainties in the $pCO_2$ in the SO (Djeutchouang et al., 2022) or examining the relationship between cyclonic and anticyclonic eddies in the South Atlantic-SO on heat and carbon (Smith et al., 2023).

Unfortunately, the computational requirements of this configuration with coupled physics-ice-BGC make sensitivity studies of long duration unrealistic. However, model-observation disagreements provided by BIOPERIANT12 suggest regions where the model configuration was poorly designed or missing processes that warrant further investigation. These will be explored with higher resolution, more appropriate datasets. Some may be addressed with newer versions of the model.

*Code and data availability.*   The model configuration is available at https://doi.org/10.5281/zenodo.13910093. Code and data used for visu-

alisations in this manuscript are available on

*Author contributions.*   Model configuration, production: NC, TCM. Analysis, visualisation, software: NC, SN, MdP, AL, TM. Writing - original draft: NC. Writing, reviewing, editing: all authors.





*Competing interests.* There are no competing interests to declare.

*Acknowledgements.* The authors acknowledge their institutional support from the CSIR Parliamentary Grant (0000005278) and the Department of Science and Innovation. SN, NC acknowledges the National Research Foundation South African National Antarctic Programme (SANAP200324510487, SANAP230503101416). PM was also funded by the European Union's Horizon 2020 research 535 and innovation programme under grant agreement No. 820989 (COMFORT). We gratefully acknowledge the Centre for High-Performance Computing (NICIS-CHPC) for providing the computational resources for the development, production, and analysis of BIOPERIANT12. NC extends appreciation to their dedicated team for their invaluable support throughout this challenging process. Additionally, NC has benefited from the expertise and support of A. Albert, J.-M. Molines, and J. Le Sommer at Laboratoire de Glaciologie et Gèophysique de l'Environnement (LGGE), Grenoble through the Marie-Curie International Research Staff Exchange Scheme, SOCCLi (The role of the Southern Ocean Carbon cycle under CLImate change, FP7-PEOPLE-2012-IRSES, 2012-2016).





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
