# Peer review of "BIOPERIANT12: a mesoscale-resolving coupled physics-biogeochemical model for the Southern Ocean"

_Geoscientific Model Development, 2024_

## Author Response (AR1)

**Author's response**

We thank the Editor and the two anonymous reviewers for their time and effort. We appreciate the comments that have helped improve the writing of the manuscript, and for pointing out where our arguments could be strengthened. The main changes are to the Introduction and Conclusions, which have been restructured. The evaluation of chlorophyll (Section 3.4. Surface chlorophyll) has been expanded to include the difficulties of evaluating this variable when both model and data are themselves inherently biased. Content has been rephrased for clarity and grammar, as indicated by the reviewers; and the use of paragraphs have been liberally applied to this version.

We have organized our response to each reviewer as follows:

1. **General Comments**
   This section addresses overarching feedback and suggestions applicable to the manuscript as a whole.

2. **Specific Comments by Section**
   Responses to individual comments are provided in relation to the following sections of the manuscript:

   - **Introduction**

   - **BIOPERIANT12 Model Configuration**

   - **Model Evaluation**

   - **Conclusions**

**Reviewer 1**

**General Comments**

**Comment:** *Overall the manuscript gives a valuable advance in modelling the Southern Ocean but the author needs to improve the writing before I can endorse.*
**Response:** I cannot disagree.
**Changes to manuscript:** The manuscript has been overhauled for clarity and grammar, whilst also following the suggestions and comments listed by both reviewers. These changes are further detailed in the responses below.

**Comment:** *I would also suggest to put the references in a different color.*
**Response:** Formatting is performed by the provided journal style file.
**Changes to manuscript:** Unchanged.

**Introduction**

**Comment:** *Even if the scientific knowledge of the article is good the introduction needs to be re-written considering that is now just one paragraph.*
**Response:** Agree.
**Changes to manuscript:** The Introduction has been parsed, paragraphs reordered and text edited. The introduction is now ordered as follows:
- The importance of Southern Ocean (SO) carbon dioxide ($CO_2$) to the global carbon cycle
- The challenges in modelling and observing the SO
- The drivers of SO variability (mesoscale dynamics)
- How models fail to capture these dynamics (seasonal cycle)
- Why models fail to capture them, i.e. computational trade-offs
- Introduction to our model, and comparative models

**BIOPERIANT12 model configuration**

**Comment:** *I think the initial condition section needs to be expanded.*
**Response:** The technical description of the initial conditions has now been expanded to include a broader context of the ocean state at the beginning of the experiment and the biogeochemistry data used for initialisation.
**Changes to manuscript:** Added L119-127.
"Biogeochemical tracers are initialised from coarse-resolution observational climatologies which provide realistic large-scale distributions. Dissolved inorganic carbon (DIC), total alkalinity (TA) are

obtained from the GLODAP annual mean climatology (Key et al., 2004); while nutrients including nitrate (NO3), phosphate (PO4), silicate (Si) and oxygen (O2) are taken from the January monthly climatology of WOA09 (Garcia et al., 2013, 2010). Dissolved organic carbon and iron fields are inherited from the BIOPERIANT05-GAA95b simulation due to lack of climatological dataset and particularly the importance of iron to the region. BIOPERIANT05-GAA95b was initialised with the same biogeochemical tracers as above and with remaining tracers from global model initial conditions (ORCA2, NEMO Consortium, 2020). Its output provides an internally consistent distribution of the aforementioned fields for the SO and is thus also used for boundary conditions. The remaining tracers in PISCES are initialised with uniform values as per standard model defaults.”

**Comment:** *In line 122, can you reference where the numbers come from?*
**Response:** The values are from default ORCA model configuration namelists.
**Changes to manuscript:** Reference added L159-161.
“For background subgrid-scale mixing, vertical eddy viscosity and diffusivity coefficients are set to $1.2 \times 10^{-4}$ m$^2$ s$^{-1}$ and $1.2 \times 10^{-5}$ m$^2$ s$^{-1}$ from default configuration namelists, e.g. ORCA2 (NEMO Consortium, 2020).”

**Model Evaluation**

**Comment:** *In line 232, do you think if you would run the model for longer than 10 years it would be less stable?*
**Response:** We did consider the effect of a longer run duration, however we did not include this in the manuscript but addressed in alongside comment in interactive discussions: “if the model were to be run over a longer duration, the model would be forced with increasing atmospheric $CO_2$ and  radiation/temperature and a trend at the surface would become more apparent. While I consider the model to be still stable if run for a further decade (I expect/hope no spurious mesoscale instabilities will develop), I cannot account for potential drift in the ocean interior and deeper layers which may have time to develop, altering the vertical water column thus affecting the surface from below.“
**Changes to manuscript:** Unchanged.

**Conclusions**

**Comment:** *In 425, in the article you have used two phytoplankton classes. IS there a reason why you choose only two classes?*
**Response:** Only 2 phytoplankton classes are available in the standard distribution of the PISCES model code. A 3rd class is available in an extended PISCES Quota model but this was not used. Addressed alongside comments in interactive discussions. “For this reference run, two phytoplankton classes were provided with the PISCES NEMO v3.4 version used. Future runs/sensitivity experiments can address this choice.”
**Changes to manuscript:** Added a description of PISCES to the model introduction L63-66.
“The PISCES biogeochemical model simulates 24 evolving prognostic tracers for carbon and nutrients cycles, as well as marine productivity, with two phytoplankton 65 groups

(nanophytoplankton and diatoms) and two zooplankton groups (microzooplankton and mesozooplankton) (Aumont and Bopp, 2006; Aumont et al., 2015)."

**Reviewer 2**

**General Comments**

**Comment:** *One of my main concerns with the manuscript is the description of its purpose. The conclusion seems to focus on the limitations of the model and on improvements needed, while arguing that this novel setup can be used for future studies. This section would benefit from a comparison of what this model improves relative to other SO models, and of what type of studies it would be suited for.*
**Response:** Agree, the above points are helpful in suggesting how the Introduction and Conclusion need strengthening.
**Changes to manuscript:** Added to the Introduction
L61-75.
"In this paper, we present our regional SO model configuration of a laterally unconstrained Antarctic Circumpolar Current (ACC) with resolved eddies and a prescribed atmosphere. BIOPERIANT12 is a regional, circumpolar, mesoscale-resolving (1/12°), contemporary ocean–ice–BGC model configuration using NEMO–PISCES.

...

This setup allows us to examine: the seasonal cycle of physical and biogeochemical processes in the surface ocean, the interface across which atmosphere–ocean exchange occurs; model–observation biases through comparison with in situ data; and simulation development through applications such as downscaling, submesoscale experiments, and sensitivity studies.

More specifically, BIOPERIANT12 allows the investigation of how sub-seasonal to synoptic-scale atmospheric forcing, such as storms, modulates seasonal buoyancy fluxes in an eddying ocean, and how these interactions shape BGC and carbon fluxes. Observational campaigns, including those with gliders, show storm events are key drivers of biological variability by influencing processes such as vertical mixing, nutrient supply, and light availability (Nicholson et al., 2022; Toolsee et al., 2024; du Plessis et al., 2022; Swart et al., 2012), yet these processes are not resolved in most ESMs. BIOPERIANT12's mesoscale-resolving resolution makes it well-suited to investigate these biophysical processes."
L86-88.
"While these configurations offer valuable insights into SO dynamics and BGC, their design choices reflect specific research objectives and therefore make them less suited for addressing processes and model representation of SO BGC coupled to realistic mesoscale dynamics that drive the observed variability and air–sea exchange."

**Comment:** *a more in-depth discussion of solutions to current biases and mismatches with observations, as well as recommendations for observations would add value to the manuscript.*

**Response:** Yes, we can better emphasise how the model can be used to highlight biases and improve mismatches with observations. Particularly, as we hope this configuration can be used to address these biases, and relates to the previous comment of the model purpose being poorly described. Addressing biases in model BGC are part of another paper (Mashifane,et al., in progress). Although we can not add recommendations for observations in this paper, we have provided some context and suggested reading.

**Changes to manuscript:** Conclusion expanded in an attempt to address comment. L506-517.

"While this model description paper focuses on introducing the BIOPERIANT12 platform and evaluating its large-scale physical and biogeochemical patterns, the use of low-resolution gridded observational datasets inherently limits the depth to which model–data mismatches can be interpreted. In high-resolution configurations such as BIOPERIANT12, discrepancies with observations often reveal limitations, not only in the model formulation but also in the observing systems themselves. BIOPERIANT12 offers an opportunity to diagnose and potentially reduce structural biases; however, doing so requires an equally robust understanding of observational uncertainty.

For physical processes, this has been discussed in the context of the development and constraining of ocean circulation models by Fox-Kemper et al. (2019). In the context of BGC, particularly for surface carbon fluxes, understanding the drivers of primary production and bloom dynamics is critical (Thomalla et al., 2023). Although satellite products have the necessary spatiotemporal coverage to inform these processes, there are limitations as addressed by Clow et al. (2024). For surface ocean pCO2, the biases and sampling limitations are discussed by Djeutchouang et al. (2022), highlighting the challenges of sparse coverage."

**Comment:** *L53: "run duration which relies on periods of sufficient observations" seems vague*

**Response:** Agree. The statement was removed as B-SOSE model runs are available for over a decade of model years.

**Changes to manuscript:** Sentence changed. L78-80.

"Another example, the B-SOSE (Biogeochemical Southern Ocean State Estimate), balances the model complexity of BGC data assimilation with coarser horizontal resolutions (1/3°, Verdy and Mazloff, 2017, and 1/6°, http://sose.ucsd.edu/)."

**Comment:** *L54-58: Sentence is too long and a bit confusing*

**Response:** Agree.

**Changes to manuscript:** Paragraph restructured and rephrased. L55-59.

"For simulations focused on SO $CO_2$ and heat fluxes, resolution is critical: it defines the spatial scales over which ocean dynamics operate to distribute BGC tracers, and significantly contributes to model–observation discrepancies. For example, mesoscale modulation of the MLD affects light and iron availability at the surface, which in turn influences phytoplankton growth and the strength of the biological carbon pump (Song et al., 2018)."

**Comment:** *L60-61: matters more for simulations of hundreds of years, but does it matter at this scale? If not, why bring this up?*

**Response:** This is a possible source of model-data mismatch that we acknowledge may occur and is a consideration when designing model experiments.

**Changes to manuscript:** In the changes to the structure of the Introduction, this sentiment is possibly clearer. Sentence changes to L59-60.

"In addition, low-resolution models are more prone to cumulative errors in BGC fields, such as nutrient and iron pools, which can propagate and amplify over time (Seferian et al., 2013)."

**BIOPERIANT12 model configuration**

**Comment:** *L75-76: "although more recent versions were available" – for which components?*
**Response:** NEMO v3.6 was the stable version at the time.
**Changes to manuscript:** Updated to be more specific, L99-101.

"Although NEMO version 3.6 was available at the time of production, version 3.4 was retained for consistency with the configuration development workflow, which involved a hierarchy of tests across increasing resolutions used to evaluate model parameters."

**Comment L79:** what is the resolution of the eddy-permitting?
**Response:** Agree, it is not obvious.
**Changes to manuscript:** Specified resolution L102-103.

"BIOPERIANT05-GAA95b, an eddy-permitting 1/2° SO biogeochemical configuration"

**Comment: L96-97:** maybe add some references for how well inputs extracted from ORCA05 are represented in that run?
**Response:** We only have references for the evaluation of pCO2 from BIOPERIANT05-GAA95b in which the variability was found to be too low compared to observations (Gregor et al., 2017). This, however, cannot be attributed to the low resolution of the model or from poor performance of the model. For both BIOPERIANT05-GAA95b and BIOPERIANT12, we expect the BGC concentrations and distribution to be governed by vertical processes and advection dominated by the strong Antarctic Circumpolar Current features rather than from input of BGC climatological boundary conditions.
Reference: Gregor, L., Kok, S., and Monteiro, P. M. S.: Empirical methods for the estimation of Southern Ocean $CO_2$ : support vector and random forest regression, Biogeosciences, 14, 5551–5569, https://doi.org/10.5194/bg-14-5551-2017, 2017.
**Changes to manuscript:** Unchanged.

**Comment:** *L103-104: I think you could expand a bit here. What biases were observed? My initial thought was that using a "normal year" would also introduce errors, so why one versus the other?*
**Response:** The seasonal cycle of $CO_2$ in the low resolution model was out of phase with observation-based datasets. This was partially explained by the seasonal ocean dynamics which led to poor representation of the observed vertical distribution of DIC (magnitude and gradients) which influenced $CO_2$ flux.
**Changes to manuscript:** Sentence changed, L133-136.

"Rather than using the full interannual series, which in coarse-resolution models can result in biases in the seasonal and vertical structure of DIC, and hence in the simulated seasonal cycle of carbon (Mongwe et al., 2016), a climatological "normal year" boundary forcing was constructed. This consisted of 5 day averaged fields computed over the period 1995–2009."

**Comment L116:** *Is this shown somewhere, to inform users?*
**Response:** We will plot and add images to the supplementary.
**Changes to manuscript:** Reference to new figures added L150-153.

"While the focus of the configuration is on upper ocean processes, particularly in the top 1000 m, it is noted that a gradual drift in deep ocean temperature (below 400 m) becomes apparent from around 2002 onward (Fig. S1e–g). This drift does not impact the surface dynamics or the primary objectives of the study, but should be considered when using the model output for investigations involving deep ocean processes."

Supplementary figure S1, updated to include temperature evolution.

[Figure]

**Figure S1.** Evolution of the BIOPERIANT12 configuration **(a)** 5–day mean, domain-averaged, surface EKE ($cm^2s^{-2}$); **(b)** zonal transport (Sv) through the Drake Passage, monthly and annual means shown; **(c)** domain-mean ocean heat content (OHC) for 0–400 m and 400-700 m. Evolution of domain-mean model temperature at **(d)** surface, 5 m and 100 m; **(e)** 200 m, 400 m, **(f)** 700 m, 1000 m, and **(g)** 2000 m and 3000 m.

**Model Evaluation**

**Comment** *L147-148: Maybe a sentence on how mean transport was calculated?*
**Response:** Done.
**Changes to manuscript:** Calculation added, L190-192.

"Transport of the ACC through the 190 Drake Passage was calculated by integrating the model's zonal velocity from surface to bottom across 69° W. The time evolution of transport (Fig. S1b), shows that BIOPERIANT12 is stable after spin-up, with an annual mean transport through the Drake Passage from 2000–2009 at 145.25 ± 5.66 Sv."

**Comment:** *L163: It's hard to see where 36-43˚S is, would be helpful to add latitudes to the maps*
**Response:** Agree.
**Changes to manuscript:** Added grid lines every 30° longitude. Added latitude and longitude labels to Figure 1 Bathymetry. Update all maps with certain labels as shown in example below.

[Figure]

**Comment:** *L169: Do you mean higher resolution in the model compared to observations?*
**Response:** Correct, thanks.
**Changes to manuscript:** Rephrased, L214-216.

"In contrast, the regional model MOMSO (Dietze et al., 2020) overestimates EKE, which is attributed to the lower resolution of the observational dataset used for comparison."

**Comment:** *L179-183: Sentence is too long*
**Response:** Agree.
**Changes to manuscript:** Rephrased, L225-230.

"Latitudinal shifts in the positions of these fronts can lead to local changes which affect heat and carbon fluxes and are thus used as a SO model evaluation metric by Russell et al. (2018).
The positions of the SubAntarctic Front (SAF) and PF are chosen to represent the northern boundary and the central ACC, respectively. Following Russell et al. (2018), we apply a simplified subsurface temperature criterion, consistent with (Orsi et al., 1995), to identify these fronts: the SAF is defined by

the 4 °C isotherm at 400 m and the PF by the 2 °C isotherm in the upper 200 m. This approach allows easy inter-model and model–observation comparisons."

**Comment:** *L185: Do you mean monthly mean for the 2000-2009 period?*
**Response:** Yes!
**Changes to manuscript:** Changed L231.
"In Fig. 3, we present the annual mean position and standard deviation of the SAF and PF in BIOPERIANT12 calculated from monthly means for the 2000–2009 period."

**Comment L200-204:** *I'm confused about the message on the final sentences. Does the variability of the front position complicate the analysis or improve bgc representation?*
**Response:** Agree, it is not clear. While contributing to model-observation discrepancies in frontal position, the rich dynamics could improve BGC representation.
**Changes to manuscript:** Rephrased L248-253.
"The high variability in frontal positioning driven by eddies in BIOPERIANT12 can lead to inconsistencies with observations, particularly in regions, such as the Kerguelen Plateau, where frontal locations are more strongly influenced by eddy activity than by the more stable meandering of coherent jets (Shao et al., 2015). Nonetheless, the model's improved representation of mesoscale processes is expected to support more realistic exchange of water masses and biogeochemical properties (Rosso et al., 2020). While this complexity complicates direct model–observation comparison of frontal positions. The use of fronts remains valuable for delineating regions for analysis."

**Comment:** Figure 3: grey shadings are all the same, and should specify which satellite
**Response:** True. Degrees of grey shading are not necessary, it's to help the reader relate the fronts to major bathymetry and not used for diagnosis, will change caption to be more general. I used the same label as Russell et al (2008) from which this plot is inspired. I will add the satellite information in the caption.
**Changes to manuscript:**
Include satellites in Table 1 "Satellite AMSR-E, AMSR-2, WindSat".
Replot Figure 3 and edit caption.
"Annual mean latitudinal position of the SubAntarctic Front (dashed line) and Polar Front (solid line) derived from temperature in BIOPERIANT12 (2000–2009), WOA13, the dataset of Orsi et al. (1995), and satellite SST Polar Front (2002–2009) from Freeman and Lovenduski (2016). Colour shaded regions are the standard deviation of the front using monthly mean temperatures. Grey shaded regions show bathymetry shallower than 3000 m."

**Comment:** *L238: area-weighing of which dataset? WOA?*
**Response:** Yes area-weighting is applied to both model and WOA but in this context I have made it confusing.
**Changes to manuscript:** Removed reference to area-weighting.

**Comment:** *L239-240: Reasoning is not super clear to me.*
**Response:** Same.
**Changes to manuscript:** Rephrased, L289-294.
"Following initialisation with climatological fields and spin-up, upper ocean temperatures in the model (upper 200 m) exhibit no significant drift over the simulation period (Fig. S1c–d), even without

the use of surface temperature restoring. Combined with the stable surface EKE discussed earlier, this suggests that surface ocean dynamics in BIOPERIANT12 remain stable throughout the analysis period. Given the consistency of model–observation biases, we do not attribute discrepancies in the physical or biogeochemical fields to spurious or transient model behaviour. Rather, remaining biases likely reflect systematic differences tied to consistent model behaviour or specific model design choices."

**Comment:** *Section 3.1.5: could the negative SST bias and positive sub-surface temperature bias be influencing MLD?*
**Response:** Interesting. The MLD is a response to the model dynamics. While SST and subsurface temperature biases may influence the MLD, simulated MLD variability and the biogeochemical response to them would be internally consistent allowing model results to be meaningfully analysed within the model framework.
**Changes to manuscript:** Unchanged.

**Comment:** *L265-267: Need more information on the calculations*
**Response:** Some detail was provided but it was not clear. Rephrasing and further information will be added to the Supplementary.
**Changes to manuscript:** Rephrased, L318-322.

"Biomes are derived from climatological fields of sea surface temperature (SST), sea ice fraction, spring/summer chlorophyll-a, and maximum mixed layer depth (MLD), using criteria defined in Fay and McKinley (2014, Table 1). SST and sea ice fraction criteria are used to distinguish between ice-covered, subpolar, and subtropical zones, while chlorophyll-a and MLD criteria reflect environmental controls on biological production, such as vertical mixing, stratification, and seasonality (i.e. permanent vs. seasonal stratification)."

**Comment:** *L271: What does SP-STPS, SA-STPS and IND-STPS stand for?*
**Response:** This was overlooked, it was defined in figure caption and not in text.
**Changes to manuscript:** Definitions added and content edited, L323-326.

"Within the BIOPERIANT12 domain, the following biomes are identified (Fig. 6): in the SO, the ice biome (SO-ICE), the subpolar seasonally stratified biome (SO-SPSS), and the subtropical seasonally stratified biome (SO-STSS); further north, the subtropical permanently stratified biomes of the South Pacific (SP-STPS), South Atlantic (SA-STPS), and Indian Ocean (IND-STPS)."

**Comment:** *L325: would be useful to have a more in-depth description of the data product for readers to interpret the model-data differences*
**Response:** Agree.
**Changes to manuscript:** Rephrased and data description added, L371-372.

"CSIR-ML6 is a gridded 1° x 1° machine learning-based reconstruction of surface ocean $pCO_2$, derived from Surface Ocean $CO_2$ Atlas (SOCAT, Bakker et al., 2016) observations and satellite-based environmental predictors (Gregor et al., 2019)."

**Comment:** *L[3]39-[3]43: Might want to rephrase it. At some point it becomes unclear if you are talking about ICE or the whole domain*
**Response:** Agree.
**Changes to manuscript:** Paragraph rewritten, L458-464.

"BIOPERIANT12 broadly captures the spatial patterns of surface chlorophyll concentrations (Fig. 11). For example, the model captures enhanced chlorophyll levels near continental margins and in frontal regions such as the Subantarctic Zone (SAZ; not shown), while lower chlorophyll concentrations are simulated in more oligotrophic regions like the South Pacific sector of the SO. However, there are notable differences in the summer pattern (Fig. 11a–c), such as the overestimation of the spatial extent of elevated chlorophyll concentrations associated with shallow topography. While the overestimation of chlorophyll occurs regardless of spatial aggregation, the inclusion of these elevated regions within biome definitions contributes to the overestimation of biome-mean values and results in variability more than twice that observed (RSD > 2.00 in Table 3)."

**Comment L356:** *Avoid repeating thus?*
**Response:** Good idea.
**Changes to manuscript:** Rephrased, L415-417.

"Dissolved iron (dFe) limits phytoplankton growth across the surface of the SO, impacting Dissolved iron (dFe) limits phytoplankton growth across the surface of the SO, impacting the functioning of marine ecosystems and, consequently, the carbon cycle. It is therefore imperative that models accurately represent the spatial and seasonal distribution of dFe."

**Comment:** *L363-365: Doesn't that depend on dFE sources, since it will not stay in the surface long? Are you talking about biases in the initialization files? I would assume this has more to do with model setup than with the choice of model*
**Response:** While model inputs (sources of iron) and initial profiles and distribution are definitely a factor, here, we are referring to the oversimplification of biogeochemical cycling which has been noted in the PISCES model. This contributes to model-data discrepancies in addition to the role played by ocean conditions.
**Changes to manuscript:** Rephrase and include more specific references, L422-425.

"Overall, the simulated open ocean surface dFe range lies on the lower end of the observed spectrum; this is consistent with previous findings that PISCES tends to underestimate open ocean dFe in the SO. This is suggested to arise from the simplification of the biological processes in the model that affects iron cycling and supply (Aumont et al., 2015; Tagliabue et al., 2016; Nicholson et al., 2019).

**Comment:** *L381-381: respectively, after the values?*
**Response:** Yes, could be less clumsy.
**Changes to manuscript:** Rephrased, L441-444.

"However, simulated surface Si is less well represented (RI = 1.79 for both SO-SPSS and SO-STSS): in the SO-SPSS, the model mean Si concentration (26.46 mmol $l^{-1}$) is substantially higher than observed (15.85 mmol $l^{-1}$), with a reduced seasonal amplitude (5.47 vs. 11.3 mmol $l^{-1}$) and only half the variability (model RSD = 0.49)."

**Comment:** *L382: How is the representation of simulated silicate dictated by laboratory experiments? This sentence was a bit confusing to me.*
**Response:** True. The treatment of silica dissolution is formulated in two phases as dictated from a laboratory experiment. However, this process may not be representative as the study received some scrutiny.
**Changes to manuscript:** Removed unnecessary detail. Rephrased, L444-445.

"These discrepancies may stem from uncertainties in the silica dissolution process and its formulation in the model Aumont et al. (2015), which may, in turn, affect the simulated diatom distribution."

**Comment:** *Figure 11: There are substantial differences in the summer pattern that need to be discussed*

**Response:** Yes, thank you for pointing this out. Not enough emphasis was given to the differences in this section, it was only discussed in the Conclusion. A more general discussion on the difficulties of direct model to satellite comparisons are now included. An in-depth look into chlorophyll and primary productivity in model versus satellite, specific to NEMO-PISCES, are being addressed in another paper.

**Changes to manuscript:** I have moved chlorophyll discussion out of the Conclusion and into Section 3.4.3 Surface chlorophyll. Paragraph added, L475-483.

"These results demonstrate that while BIOPERIANT12 captures key spatial and seasonal features of chlorophyll in the SO, it likely underrepresents the higher-frequency biological variability evident in observations; even at mesoscale-resolving resolution. Underlying ocean dynamics that could be further improved in BIOPERIANT12 include the representation of shelf-slope dynamics and mixing processes. For instance, enhanced vertical mixing may prolong bloom duration, as seen in the SO-SPSS biome, where the modelled bloom persists nearly three months beyond satellite-based estimates. Additionally, the biogeochemical processes that drive productivity and blooms in the SO such as nutrient limitation (especially iron), fixed phytoplankton stoichiometry, and prescribed chlorophyll-to-carbon ratios, also require further refinement. The overestimation of chlorophyll magnitude and mismatches in bloom timing suggest that processes governing phytoplankton bloom dynamics, particularly those modulated by mesoscale physical variability, warrant further investigation."

**Comment:** *L396-397:wouldn't the overestimation be true regardless of biome classification?*

**Response:** Correct, I was trying to incorporate the insensitivity of biome aggregation.

**Changes to manuscript:** Rephrased, L 461-464.

"While the overestimation of chlorophyll occurs regardless of spatial aggregation, the inclusion of these elevated regions within biome definitions contributes to the overestimation of biome-mean values and results in variability more than twice that observed (RSD > 2.00 in Table 3)."

**Comment:** *Section 3.4.3: How good is the satellite temporal coverage in the different biomes? Is it properly representing intra seasonal to interannual variability? Discussing potential biases in the observation might help the discussion here*

**Response:** Thanks for this point and excellent suggestion. Similar to the Carbon data product, more information should be supplied in line.

**Changes to manuscript:** Added, L455-468.

"To evaluate the model's representation of biological variability, we compare surface chlorophyll concentrations from BIOPERIANT12 with the OC-CCI v6 satellite-derived dataset (Table 1) gridded at a comparable resolution of 9 km and aggregated weekly to match the model output. BIOPERIANT12 broadly captures the spatial patterns of surface chlorophyll concentrations (Fig. 11)."

Also added sources of biases, L484-488.

"However, chlorophyll model vs. satellite data comparisons are inherently challenging, with both sources subject to internal biases and uncertainties. Limitations in satellite observations arise from the satellite itself, such as solar zenith angle, cloud cover, and sea ice contamination, particularly at high latitudes; as well as from environmental complexities of the SO like the presence of subsurface chlorophyll maxima and algorithms not well suited to these conditions (Clow et al., 2024; Aumont et al., 2015)."